# AgentRL: Scaling Agentic Reinforcement Learning with a Multi-Turn, Multi-Task Framework

## Abstract

Recent advances in large language models (LLMs) have sparked growing interest in building generalist agents that can learn through online interactions. However, applying reinforcement learning (RL) to train LLM agents in multi-turn, multi-task settings remains challenging due to lack of scalable infrastructure and stable training algorithms. In this work, we present the AgentRL framework for scalable multi-turn, multi-task agentic RL training. On the infrastructure side, AgentRL features a fully-asynchronous generation-training pipeline for efficient multi-turn RL. To support heterogeneous environment development in multi-task RL, we design a unified function-call based API interface, containerized environment development, and a centralized controller. On the algorithm side, we propose cross-policy sampling to encourage model exploration in multi-turn settings and task advantage normalization to stabilize multi-task training. Experiments show that AgentRL, trained on open LLMs across five agentic tasks, significantly outperforms GPT-5, Clause-Sonnet-4, DeepSeek-R1, and other open-source LLM agents. Multi-task training with AgentRL matches the best results among all task-specific models. AgentRL is open-sourced at `https://anonymous.4open.science/r/AgentRL-ICLR-C351.` and has also been adopted for developing other open-source LLM agents.

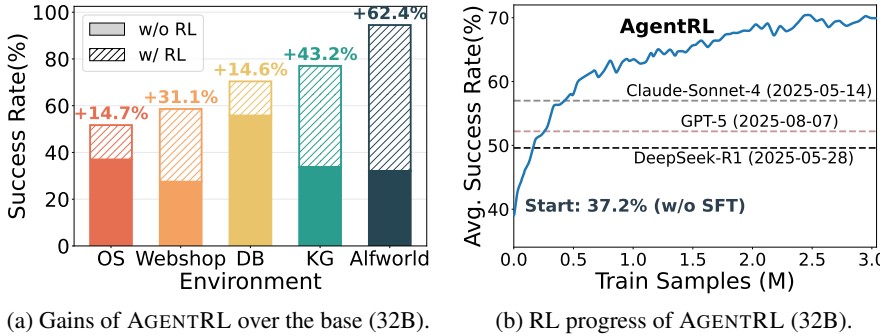

(a) Gains of AgentRL over the base (32B).  (b) RL progress of AgentRL (32B).

Figure 1: Overall performance of AgentRL.

## 1 Introduction

Reinforcement learning (RL) trains an agent to act by interacting with an environment and optimizing its policy to maximize cumulative rewards. This principle has been effectively adapted for large language models (LLMs) through reinforcement learning from human feedback (RLHF) (Ouyang et al., 2022; OpenAI, 2022), where the LLM itself acts as the agent and its policy is refined based on feedback from a learned reward model. This optimization process, typically based on proximal policy optimization (PPO) (Schulman et al., 2017), aligns the model's outputs with desired behaviors.

More recently, reinforcement learning with verifiable rewards (RLVR) (Shao et al., 2024) has extended RL to reasoning tasks. Instead of relying on a learned reward model, RLVR uses automatically verifiable signals, such as correctness checks in math or unit tests in code. This shift to objective rewards enables significant simplification of the algorithmic design. For example, the group relative

Table 1: AGENTRL vs. other RL frameworks and methods. Interactive Envs: real-time interaction with the environment during training; Heterogeneous Envs: training with diverse environments.

| Method | Agentic Setting | | Infrastructure | | |
|---|---|---|---|---|---|
| | Multi-Turn | Multi-Task | Full-Async | Interative Envs | Heterogeneous Envs |
| VeRL (Sheng et al., 2024) | ✗ | ✗ | ✗ | ✗ | ✗ |
| OpenRLHF (Hu et al., 2024) | ✗ | ✗ | ✗ | ✗ | ✗ |
| NeMo-Aligner (Shen et al., 2024) | ✗ | ✗ | ✗ | ✗ | ✗ |
| AReaL (Fu et al., 2025) | ✓ | ✗ | ✓ | ✗ | ✗ |
| AgentTuning (Zeng et al., 2024) | ✓ | ✓ | ✗ | ✗ | ✗ |
| EasyR1 (Zheng et al., 2025a) | ✗ | ✗ | ✗ | ✗ | ✗ |
| DigiRL (Bai et al., 2024) | ✓ | ✗ | ✗ | ✓ | ✗ |
| RAGEN (Wang et al., 2025b) | ✓ | ✗ | ✗ | ✓ | ✗ |
| ToolRL (Qian et al., 2025) | ✗ | ✗ | ✗ | ✗ | ✗ |
| GiGPO (Feng et al., 2025) | ✓ | ✗ | ✗ | ✓ | ✗ |
| ARPO (Lu et al., 2025a) | ✓ | ✗ | ✗ | ✓ | ✗ |
| **AGENTRL (ours)** | ✓ | ✓ | ✓ | ✓ | ✓ |

policy optimization (GRPO) (Shao et al., 2024) algorithm further simplifies PPO and improves LLMs' RL training efficiency. Recent LLMs leveraging RLVR—e.g., DeepSeek-R1 (DeepSeek-AI et al., 2025) and T1 (Hou et al., 2025)—have achieved strong performance in reasoning.

However, these RL for LLM achievements have been largely limited to *single-turn* settings for a *single task*, where an agent interacts with the given environment only once for feedback (Qi et al., 2024; Bai et al., 2024; Zheng et al., 2025b; Feng et al., 2025; Qian et al., 2025; Yue et al., 2023). First, to solve agentic tasks with *multi-turn* settings (OpenAI, 2025c; Jin et al., 2025; Lu et al., 2025a; Feng et al., 2025; Lu et al., 2025b), the agent must collect feedback through dynamic interactions with environments (Deng et al., 2023; Wei et al., 2025). In this case, the LLM is trained as an autonomous agent that performs multi-turn reasoning, interacts with tools or environments, and adapts its behavior over extended trajectories, that is, the problem of agentic RL. Second, building a generalist agent that can handle *diverse tasks* has long been a goal for RL. Scaling to heterogeneous multi-task environments in multi-turn settings for agentic RL requires advances in both LLM training infrastructure and algorithm design. Table 1 lists existing solutions.

In this work, we present a multi-turn, multi-task framework AGENTRL to scale agentic RL training. AGENTRL includes RL infrastructure, environment, and algorithm designs to address the challenges listed in Table 2. On the infrastructure side, we implement an asynchronous generation-training pipeline that can reduce GPU idle bubbles and improve multi-turn training efficiency. On the environment side, we develop a scalable environment deployment infrastructure with a unified function-call based API interface, containerized deployment, and centralized controller to manage the lifecycle of thousands of parallel training episodes. To further support heterogeneous environment scaling, we introduce consistent interfaces at the controller level. On the algorithm side, we present the cross-policy sampling strategy to encourage model exploration that is negatively impacted by the large state space in the multi-turn setting. We also introduce task advantage normalization to mitigate the training instability resulting from the heterogeneity in different tasks.

We apply AGENTRL on open LLMs—Qwen2.5 (Qwen et al., 2025) and GLM-4-9B (GLM et al., 2024)—across five agentic tasks: ALFWorld, DB, KG, OS, and Webshop (Shridhar et al., 2021; Yao et al., 2022; Liu et al., 2024c). Experiments show that AGENTRL achieves state-of-the-art results, significantly outperforming GPT-5 (OpenAI, 2025a), Claude-Sonnet-4 (Anthropic, 2025) and DeepSeek-R1 (DeepSeek-AI et al., 2025) (Figure 1). The single model trained with five tasks together can match the best performance of five models trained separately for individual tasks, while also generalizing into unseen tasks, e.g., BFCL-v3 (Patil et al., 2025). Finally, extensive ablations demonstrate that the algorithmic design choices in AGENTRL bring consistent performance benefits.

The contributions of this work are summarized as follows:

- We develop an asynchronous, multi-task framework AGENTRL for scalable agentic RL training and robust heterogeneous environment deployment.
- We design a cross-policy sampling strategy to encourage exploration in multi-turn settings and task advantage normalization to stabilize multi-task RL training.
- AGENTRL achieves state-of-the-art results on various LLM agent tasks, with promising generalization to unseen tasks, demonstrating the potential of building a generalist LLM agent.

Table 2: Challenges in agentic RL compared to single-turn RL

|  | Infrastructure | Algorithm |
|---|---|---|
| **Single-Turn** | synchronous rollouts | stable and scalable training |
| **Multi-Turn** | compute inefficiency in synchronous rollouts, requiring asynchronous training; difficulty in scaling interactive homogeneous environments | multi-turn tasks demand greater exploration due to larger state spaces, but exploration declines during training |
| **Multi-Task** | difficulty in unifying heterogeneous environments | performance drop from task interference and lack of generalization |

## 2 THE AGENTIC RL PROBLEM AND ITS CHALLENGES

The shift from single-turn to multi-turn defines the problem of agentic RL, where the LLM acts as an autonomous agent that performs multi-turn reasoning, interacts with tools or environments, and adapts its behavior over extended trajectories. Formally, this can be formulated as a Markov Decision Process(MDP) (Puterman, 2014), a tuple $(\mathcal{S}, \mathcal{A}, P, r, \rho)$, where $\mathcal{S}$ is the state set, $\mathcal{A}$ the action set, $P$ the state-transition probability, $r$ the reward function, and $\rho$ the initial state distribution. In a single-step case, $P$ is trivial and the problem reduces to a multi-armed bandit. In contrast, multi-step MDPs involve non-trivial state evolution over multiple transitions. The definition is listed in Appendix B.

Moreover, most LLM agents have focused on training a separate policy for each individual task (Zheng et al., 2025b; Feng et al., 2025; Qian et al., 2025). That means multiple LLMs have to be trained, one for each environment or task, respectively. How to build a generalist agent that can handle diverse tasks remains largely unexplored. Table 2 summarizes the challenges that go beyond single-turn RL.

**Infrastructure Challenges in Multi-Turn RL.** In the single-turn setting, RL is often run in a synchronous way with an interleaved generation-training pipeline (Hu et al., 2024; Sheng et al., 2024). For agentic tasks, generating long trajectories and frequent interactions with the environment is slow, time-consuming, and highly variable compared to single-turn scenarios. As a result, GPUs that handle short trajectories have to stay idle to wait for the generation completion of long trajectories. The imbalance significantly reduces training efficiency and prevents RL scaling, thus requiring an asynchronous RL training framework.

On the environment side, multi-turn training requires rollouts to run in an interactive environment, which places high demands on the concurrent deployment and management of a large number of homogeneous environments.

**Algorithm Challenges in Multi-Turn RL.** On the algorithm side, most existing sampling strategies are designed for single-turn settings. Improving exploration and sampling efficiency in multi-turn scenarios is therefore critical for agentic RL training.

**Infrastructure Challenges in Multi-Task RL.** By definition, multi-task RL requires an architecture that can manage diverse environments. One major challenge lies in the differences in environment interfaces, state-action representations, and computational demands. Effective and scalable integration of these environments is essential for scaling agentic training efficiently across diverse tasks.

**Algorithm Challenges in Multi-Task RL.** Most existing RL approaches focus on training a single agent task (Jin et al., 2025; Qian et al., 2025; Feng et al., 2025). Thus, developing effective methods for jointly optimizing multiple agent tasks while ensuring training stability remains an open challenge.

## 3 THE AGENTRL FRAMEWORK

In this work, we develop an agentic RL framework—AGENTRL—to support multi-turn and multi-task RL training, as shown in Figure 2. AGENTRL implements asynchronous training and environment deployment to improve efficiency in multi-turn and multi-task settings. It also introduces cross-policy sampling and task advantage normalization to stabilize the RL training. Together, these technical designs and implementations address the challenges outlined in Table 2, and thus enable the generalist agent training by scaling multiple environments.

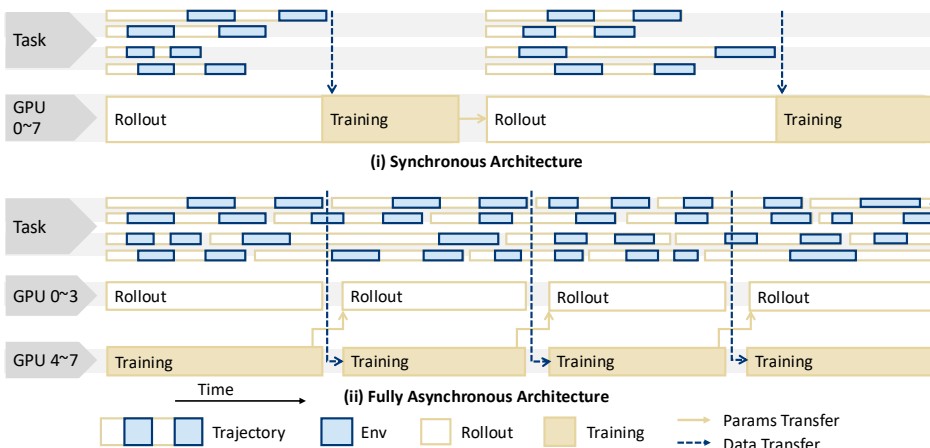

Figure 2: An overview of AGENTRL. Top: asynchronous training and rollout flows. Bottom: the environment framework where a controller manages multiple workers to provide environments, and the rollout details, including cross-policy sampling and task advantage normalization.

## 3.1 MULTI-TURN AGENTIC RL

**Asynchronous Training Framework.** To overcome the efficiency bottlenecks of synchronous batching, we introduce an asynchronous rollout-training strategy based on coroutine scheduling. The rollout engine runs in a dedicated resource group and executes asynchronously with training. The training module continuously pulls available data from the rollout engine after each update, without waiting for an entire batch of rollouts to finish. In addition, it accepts a dynamic batch size that fluctuates within a certain range. This design enables the scheduler to fill idle GPU slots with available coroutines, reducing pipeline bubbles and improving overall throughput.

Figure 3: Synchronous vs. Asynchronous Training. The asynchronous design improves efficiency by separating data rollout and model training on different resource groups.

As illustrated in Figure 3, rollout and training are decoupled. They run concurrently and communicate asynchronously. This enables efficient hardware scheduling, as shown in Figure 4, where the asynchronous pipeline in AGENTRL brings significant throughput gains over the synchronous one.

**To avoid the off-policy bias in the pipeline**, we set a maximum size of the data queue and enforce all trajectories to be moved to the training engine at each step. This ensures that data will not accumulate in the queue. In doing so, all trajectories are kept as up-to-date as possible with the latest policy, which later experiments sug-

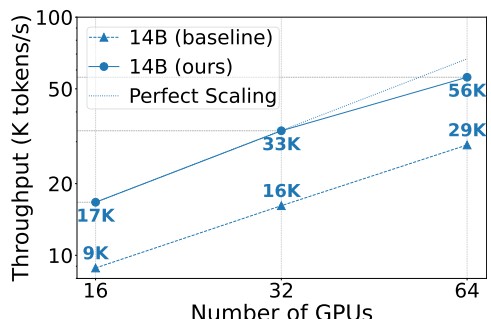

Figure 4: Throughput of AGENTRL vs. the synchronous baseline for 14B parameter (Qwen2.5) models on Webshop (log-scale for both axes).

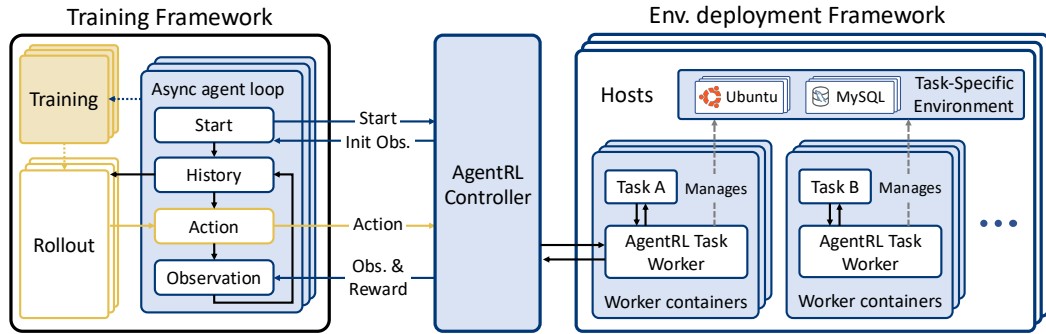

Figure 5: The AGENTRL training pipeline, decoupled into a Training Framework and an Environment Deployment Framework, organized by a central AGENTRL Controller. The Training Framework is responsible for policy rollouts and updates, while the Environment Deployment Framework manages scalable, containerized task environments that provide feedback.

gest to be acceptable. This is further discussed in Appendix B.4.

**Scalable Agentic Environment Infrastructure.** To enable large-scale agentic RL, we develop a scalable environment deployment infrastructure, shown in Figure 5. It includes the following components: *1. Function-call based environment interface.* To simplify environment interactions, we introduce a unified, function-call based API. This replaces complex custom action formats and thus enables centralized management and monitoring. *2. Containerized deployment.* Each task environment is containerized as an isolated execution unit. This design improves resource allocation, isolates faults between concurrent sessions, and supports seamless deployment on diverse hardware. *3. Centralized high-performance controller.* A central controller, acts as the global orchestrator for the training engine. It is optimized for high-concurrency workloads and manages the lifecycle of thousands of parallel training episodes.

**Cross-Policy Sampling Strategy.** During RL training, model exploration typically declines over time. This problem becomes more severe in the multi-turn setting with large state spaces. Similarly, Shumailov et al. (2024) reported that repeated training on self-generated data leads to degraded capability and reduced variance. We observed similar phenomenon in our training.

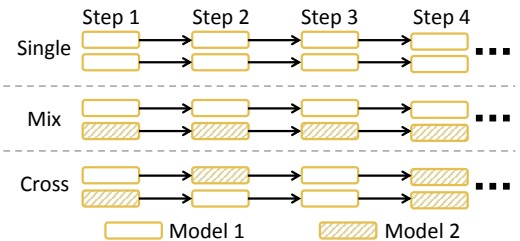

Figure 6: Different rollout strategies. In *single* model generation, all steps of all traces are generated by the same model. In *mix* mode, half of the samples are generated by each model. In *cross-policy* mode, all samples are generated with cross-policy sampling strategy.

To overcome this issue, we propose a cross-policy sampling strategy (see Figure 6), where multiple LLMs are used to generate actions within a single trajectory. The goal of aggregating data from different models is to increase the diversity of the candidate pool while preserving overall quality. Specifically, cross-policy sampling constructs trajectories by allowing actions at each step to be randomly drawn from the pool of available models, rather than committing to a single model.

Its advantage lies in that the language component of each state is still constrained to remain valid, while the expanded sampling enlarges the coverage of language states that can reach successful outcomes in the environment. By exploring paths that would not appear under any single model, cross-policy sampling increases the likelihood of visiting goal-relevant states without drifting into incoherent or invalid linguistic regions. Details can be found in Appendix B.3.

During RL training, it is hard to incorporate models with different architectures in the pipeline. Instead, we let the model do cross-policy sampling with its early version. Specifically, we mark a set of rollout engines as stale engines; these engines update parameters every multiple steps instead of one step. Early experiments verified the effect of the cross-policy sampling strategy (see Section 4.3).

## 3.2 MULTI-TASK AGENTIC RL

**Heterogeneous Environment Deployment.** Multi-task RL requires the environment deployment framework to generalize beyond a single task or environment. To host, schedule, and monitor heterogeneous environments under the same infrastructure without incurring additional integration cost, we propose to expose consistent interfaces at both the worker and controller levels. This supports AGENTRL to scale the task (environment) set in size and diversity gracefully.

We have two complementary designs: On the *environment* side, we unify the worker API across all tasks, such that each task can be instantiated and managed using an identical set of lifecycle operations. On the *training* side (Figure 5), the controller provides a single gateway API to the RL engine, abstracting away task heterogeneity and exposing multi-task execution as a transparent extension of the single-task case.

**Task Advantage Normalization.** In multi-task RL, agentic tasks often differ substantially in difficulty, sequence length, and sampling efficiency. Such heterogeneity can cause standard RL algorithms to learn at very different rates across tasks. Consequently, one task may exhibit clear reward improvements, while another shows negligible progress, leading to training instability and performance imbalance.

We normalize the token-level advantage within each to mitigate this issue. For an LLM-based policy, each high-level action $a_t$ consists of multiple tokens $\{y_{t,k}\}_{k=1}^{L_t}$. We compute token-level advantage estimates $\hat{A}_{i,s,g,t,k}$ for each token occurrence, where $i$ denotes the task index, $s$ the sample index within the task, $g$ the trajectory index within the group, $t$ the environment step, and $k$ the token position within $a_t$.

Let $\mathcal{A}_i^{\text{tok}} = \left\{ \hat{A}_{i,s,g,t,k} \;\middle|\; 1 \leq s \leq S_i,\ 1 \leq g \leq K_{i,s},\ 1 \leq t \leq T_{i,s,g},\ 1 \leq k \leq L_{i,s,g,t} \right\}$ denote the set of token-level advantages for all tokens in the current batch of task $i$, where $S_i$ is the number of samples, $K_{i,s}$ the number of trajectories per sample, $T_{i,s,g}$ the number of env steps in trajectory $\tau_{i,s,g}$, and $L_{i,s,g,t}$ the number of tokens in action $a_t$.

We normalize each token's advantage within its task batch as:

$$\tilde{A}_{i,s,g,t,k} = \frac{\hat{A}_{i,s,g,t,k} - \mu_i}{\sigma_i}, \tag{1}$$

where $\mu_i = \text{mean}(\mathcal{A}_i^{\text{tok}})$ and $\sigma_i = \text{std}(\mathcal{A}_i^{\text{tok}})$. This ensures that, for each task $i$, the distribution of token-level advantages in a batch has zero mean and unit variance, helping to reduce inter-task variance and stabilize multi-task optimization.

## 4 EXPERIMENTS

**Data.** We accommodate five agentic tasks (ALFWorld, DB, KG, OS, WebShop) (Liu et al., 2024c) to the AGENTRL infrastructure. The details of the dataset construction and unifying the function-call format are provided in Appendix C. To ensure that all tasks are sampled uniformly during training, we replicate smaller datasets such that each task appears approximately the same number of times as the largest task. Specifically, we sequentially cycle through multiple datasets, yielding one element from each in turn to produce interleaved output samples.

**Baselines.** The closed-source API-based baselines include Claude-Sonnet (Anthropic, 2025), GPT-5 (OpenAI, 2025a), and o-series models (OpenAI, 2025b). The general open models adopted include the Qwen2.5-Instruct series (14B, 32B, and 72B) (Qwen et al., 2025), DeepSeek-V3 (Liu et al., 2024a), and DeepSeek-R1 (DeepSeek-AI et al., 2025). We also compare against agent training methods on AGENTBENCH, including Hephaestus (Zhuang et al., 2025), Agent-FLAN (Chen et al., 2024b), and AgentLM (Zeng et al., 2024).

### 4.1 MAIN RESULTS

We apply AGENTRL on open models, including Qwen2.5-Instruct series and GLM-4-9B-0414. Note that there is *no warm-up supervised fine-tuning* before applying AGENTRL to all Qwen models. The main results are listed in Table 3.

Table 3: Main results (task success rate). Average and standard deviation of four repeats on each task are reported. The '*' indicates reward results directly extracted from the original papers.

| Model | ALFWorld | DB | KG | OS | Webshop | AVG |
|---|---|---|---|---|---|---|
| *API LLMs (Prompting)* | | | | | | |
| Claude-Sonnet-3.7 (2025-02-19) | $61.1_{\pm3.0}$ | $68.5_{\pm0.8}$ | $59.8_{\pm1.0}$ | $36.5_{\pm4.1}$ | $40.1_{\pm1.5}$ | 53.2 |
| Claude-Sonnet-3.7 Thinking (2025-02-19) | $54.1_{\pm3.0}$ | $68.4_{\pm0.3}$ | $38.2_{\pm2.2}$ | $53.1_{\pm1.8}$ | $36.0_{\pm1.7}$ | 50.0 |
| Claude-Sonnet-4 (2025-05-14) | $73.6_{\pm2.6}$ | $70.1_{\pm0.7}$ | $63.4_{\pm1.7}$ | $45.3_{\pm2.8}$ | $34.6_{\pm1.6}$ | 57.4 |
| Claude-Sonnet-4 Thinking (2025-05-14) | $69.0_{\pm3.2}$ | $68.4_{\pm1.0}$ | $64.4_{\pm1.9}$ | $51.0_{\pm2.3}$ | $38.3_{\pm2.8}$ | 58.2 |
| GPT-4o (2024-11-20) | $28.3_{\pm2.8}$ | $54.3_{\pm2.2}$ | $49.3_{\pm2.7}$ | $38.5_{\pm3.2}$ | $27.8_{\pm2.2}$ | 39.6 |
| o3-mini (2025-01-31) | $28.4_{\pm1.3}$ | $56.5_{\pm0.5}$ | $51.8_{\pm0.9}$ | $35.1_{\pm1.7}$ | $32.7_{\pm1.5}$ | 40.9 |
| o4-mini (2025-04-16) | $32.6_{\pm1.8}$ | $63.4_{\pm0.3}$ | $32.4_{\pm3.0}$ | $41.8_{\pm1.0}$ | $28.5_{\pm1.8}$ | 39.7 |
| GPT-5 (2025-08-07) | $65.4_{\pm2.0}$ | $63.2_{\pm0.7}$ | $64.1_{\pm1.8}$ | $34.5_{\pm1.0}$ | $33.7_{\pm2.6}$ | 52.2 |
| *Open LLMs (Prompting)* | | | | | | |
| DeepSeek-V3 (2025-03-24) | $31.9_{\pm2.0}$ | $58.4_{\pm1.2}$ | $14.0_{\pm2.0}$ | $53.0_{\pm1.0}$ | $23.4_{\pm2.5}$ | 36.1 |
| DeepSeek-R1 (2025-05-28) | $51.4_{\pm4.1}$ | $60.4_{\pm0.5}$ | $50.2_{\pm2.7}$ | $53.6_{\pm1.0}$ | $31.0_{\pm1.6}$ | 49.3 |
| Qwen2.5-14B-Instruct | $8.7_{\pm3.1}$ | $48.4_{\pm2.2}$ | $35.3_{\pm3.0}$ | $26.0_{\pm3.1}$ | $17.6_{\pm1.0}$ | 27.2 |
| Qwen2.5-32B-Instruct | $32.1_{\pm3.9}$ | $55.8_{\pm0.6}$ | $33.8_{\pm1.5}$ | $37.0_{\pm1.5}$ | $27.5_{\pm2.3}$ | 37.2 |
| Qwen2.5-72B-Instruct | $47.5_{\pm3.3}$ | $45.3_{\pm0.9}$ | $26.5_{\pm3.1}$ | $49.5_{\pm3.5}$ | $35.4_{\pm2.7}$ | 40.8 |
| *Open LLMs (Agent Training)* | | | | | | |
| Hephaestus-8B-Base | 30.0 | 32.3 | 16.0 | 20.8 | $60.5^*$ | 31.9 |
| Hephaestus-8B-IFT | 46.0 | 29.7 | 21.2 | 20.8 | $63.9^*$ | 36.3 |
| AgentLM-7B | 84.0 | 30.6 | 18.1 | 17.4 | $63.6^*$ | 42.7 |
| AgentLM-13B | 76.0 | 33.7 | 26.8 | 18.1 | $70.8^*$ | 45.1 |
| AgentLM-70B | 86.0 | 37.7 | 47.0 | 21.5 | $64.9^*$ | 51.4 |
| AGENTRL | | | | | | |
|   w/ Qwen2.5-3B-Instruct | $92.4_{\pm0.5}$ | $60.0_{\pm1.1}$ | $55.0_{\pm2.0}$ | $40.5_{\pm0.9}$ | $52.1_{\pm0.9}$ | **60.0** |
|   w/ Qwen2.5-7B-Instruct | $91.5^{\dagger}_{\pm0.9}$ | $63.7_{\pm0.5}$ | $57.8_{\pm2.3}$ | $40.8_{\pm1.2}$ | $56.1_{\pm0.6}$ | **62.0** |
|   w/ Qwen2.5-14B-Instruct | $91.5_{\pm0.9}$ | $72.2_{\pm0.9}$ | $72.8_{\pm1.8}$ | $43.6_{\pm1.9}$ | $58.5_{\pm1.2}$ | **67.7** |
|   w/ Qwen2.5-32B-Instruct | $94.5_{\pm0.5}$ | $70.4_{\pm0.5}$ | $77.0_{\pm1.2}$ | $51.7_{\pm1.8}$ | $58.6_{\pm0.9}$ | **70.4** |
|   w/ GLM-4-9B-0414 | $93.3_{\pm0.5}$ | $66.9_{\pm0.4}$ | $75.7_{\pm1.8}$ | $33.2_{\pm1.7}$ | $55.9_{\pm1.9}$ | **65.0** |

[†] We provide a one-shot demonstration for Qwen2.5-7B-Instruct in ALFWorld evaluation, as it fails to generate valid tool call format in the environment.

Table 4: Multi-Task vs. Single-Task with Qwen2.5-14B-Instruct.

| Model | ALFWorld | DB | KG | OS | Webshop | AVG |
|---|---|---|---|---|---|---|
| AGENTRL-ALFWorld | $89.7_{\pm1.6}$ | $49.7_{\pm1.6}$ | $22.3_{\pm3.1}$ | $33.7_{\pm3.1}$ | $15.9_{\pm0.5}$ | 42.3 |
| AGENTRL-DB | $0.2_{\pm0.5}$ | $73.9_{\pm0.7}$ | $26.2_{\pm1.7}$ | $43.1_{\pm1.3}$ | $16.0_{\pm0.9}$ | 31.9 |
| AGENTRL-KG | $4.6_{\pm1.1}$ | $57.6_{\pm0.8}$ | $72.2_{\pm1.5}$ | $40.3_{\pm2.4}$ | $19.5_{\pm2.0}$ | 38.8 |
| AGENTRL-OS | $5.7_{\pm1.2}$ | $58.2_{\pm1.2}$ | $25.3_{\pm1.6}$ | $39.8_{\pm1.8}$ | $22.0_{\pm2.3}$ | 30.2 |
| AGENTRL-Webshop | $0.0_{\pm0.0}$ | $57.9_{\pm2.6}$ | $30.7_{\pm2.2}$ | $40.1_{\pm0.7}$ | $60.3_{\pm1.3}$ | 37.8 |
| Best of Five Models Above | $89.7_{\pm1.6}$ | $73.9_{\pm0.7}$ | $72.2_{\pm1.5}$ | $43.1_{\pm1.3}$ | $60.3_{\pm1.3}$ | **67.8** |
| AGENTRL (One Model) | $91.5_{\pm0.9}$ | $72.2_{\pm0.9}$ | $72.8_{\pm1.8}$ | $43.6_{\pm1.9}$ | $58.5_{\pm1.2}$ | **67.7** |

**SOTA Performance.** Our AGENTRL framework achieves state-of-the-art performance across five tasks in AGENTBENCH-FC (see Appendix C), establishing a new top average success rate of 70.4%. Compared to the original Qwen2.5-Instruct models under prompting, AGENTRL yields substantial improvements, highlighting the effectiveness of reinforcement learning training. Notably, all AGENTRL-trained models, from 3B to 32B, consistently outperform strong baselines including leading models such as GPT-5, Claude-Sonnet-4 Thinking, and DeepSeek-R1.

**Multi-Task vs. Single-Task.** Table 4 shows that single-task RL agents excel only in their specific training environment but fail to generalize, yielding poor transfer across tasks. In contrast, our multi-task AGENTRL achieves nearly identical performance to the "best-of-five" single-task specialists while maintaining strong results on all tasks simultaneously. This highlights the effectiveness of multi-task training in acquiring generalizable skills without sacrificing peak performance.

Table 5: Generalization Performance on BFCL-v3.

| Model | single-turn | | multi-turn | overall |
| --- | --- | --- | --- | --- |
| | nonlive | live | | |
| Qwen2.5-32B-Instruct | $86.0_{\pm 0.2}$ | $77.4_{\pm 0.1}$ | $16.2_{\pm 0.6}$ | 59.9 |
| AGENTRL w/ Qwen2.5-Instruct-32B | $85.8_{\pm 0.2}$ ↓0.2 | $79.3_{\pm 0.2}$ ↑1.9 | $19.2_{\pm 0.8}$ ↑3.0 | 61.4 ↑1.5 |

**Generalization on BFCL-v3.** To examine generalization, we evaluate the AGENTRL model (trained on ALFWorld, DB, KG, OS, and Webshop) on the BFCL-v3 benchmark (Patil et al., 2025). BFCL-v3 evaluates the model's multi-step function calling ability. As shown in Table 5, AGENTRL demonstrates clear improvements on multi-turn tasks and modest gains on single-turn tasks. These results suggest that our approach can enhance the generalizability of function calling, providing a step toward more broadly capable agentic LLMs. This is further discussed in Appendix D.4.

Table 6: Ablation on cross-policy sampling and task advantage normalization.

| Method | AF | DB | KG | OS | WS | AVG |
| --- | --- | --- | --- | --- | --- | --- |
| AGENTRL-14B | $93.1_{\pm 0.5}$ | $64.0_{\pm 0.5}$ | $67.7_{\pm 2.0}$ | $45.1_{\pm 2.0}$ | $55.0_{\pm 0.7}$ | 65.0 |
| - cross sampling | $91.9_{\pm 1.2}$ | $61.6_{\pm 1.0}$ | $55.7_{\pm 1.4}$ | $39.7_{\pm 2.3}$ | $54.5_{\pm 1.3}$ | 60.7 |
| - task adv. norm | $91.1_{\pm 0.9}$ | $62.6_{\pm 0.7}$ | $54.7_{\pm 1.6}$ | $38.0_{\pm 2.0}$ | $50.6_{\pm 1.7}$ | 59.4 |

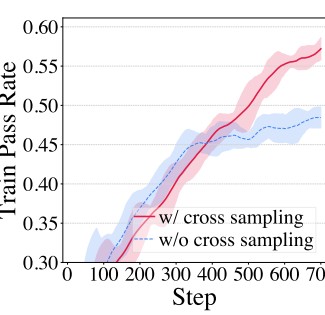

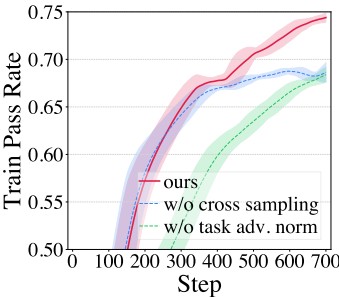

(a) Cross-Policy Sampling in KG.  (b) Task Adv. Norm. in ALFWorld.  (c) Average over 5 environments.

Figure 7: Ablation studies. (c): The combined effect of Cross-Policy Sampling and Task Advantage Normalization, averaged over five environments.

## 4.2 ABLATION STUDY

**Cross-Policy Sampling.** Table 6 suggests AGENTRL trained without cross-policy sampling performs worse. This phenomenon is especially obvious in some tasks/environments. We demonstrate the pass rate on KG during training in Figure 7a as an example; the model's capability reaches the top earlier than the model trained with cross-policy sampling. These results demonstrate that cross-policy sampling is able to explore more possible states, especially in more open-ended environments during training, thus expanding the border of the model's capability.

**Task Advantage Normalization.** Table 6 suggests that removing task advantage normalization leads to clear performance drops. Also, as shown in Figure 7b, the training efficacy is severely reduced and demonstrates fluctuations on some tasks. When removing the task advantage normalization, the model tends to learn different tasks at different rates instead of learning jointly. These results indicate that normalizing the advantage for each task effectively stabilizes multi-task training and reduces negative interference, resulting in more robust and consistent learning across tasks.

## 4.3 VERIFYING THE EFFECT OF THE CROSS-POLICY SAMPLING STRATEGY

**Applying Cross-Policy sampling in Inference.** The proposed cross-policy sampling strategy samples actions from a pool of models (as depicted in Figure 6). To verify that the cross-policy sampling strategy effectively promotes model exploration, we first directly applied our method to inference. We conducted experiments using the Qwen (Qwen et al., 2025) and Llama (Grattafiori

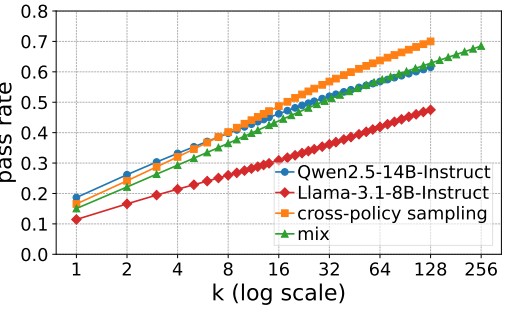 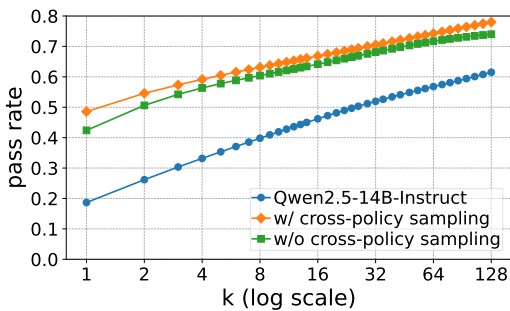

(a) Cross-policy sampling on Webshop. The *mix* strategy combines data from both models, so its maximum $K$ is twice that of the other strategies.

(b) Results from preliminary experiments on the WebShop environment. Note that settings are not completely the same as those in the main experiments.

Figure 8: Effects of cross-policy sampling in inference (a) and training (b) on Webshop.

et al., 2024) models in the WebShop (Yao et al., 2022) environment. As shown in Figure 8a, we observe that in low-$k$ regimes, the performance of the cross-policy sampling strategy is slightly lower than the best single model strategy. However, as $k$ increases, a surprising trend emerges: the cross-policy sampling strategy eventually surpasses both individual models in `pass@k` metrics. The performance of the cross-policy sampling strategy also surpasses mixing two models' trajectories, demonstrating that the strategy has effectively explored something outside both models' capability boundaries. This provides strong evidence for our theoretical analysis.

**Applying Cross-Policy sampling in RL.** To further verify the effectiveness of the cross-policy sampling strategy during RL training, we conduct a training experiment on the Webshop task. As shown in Figure 8b, both trained models demonstrated a significant improvement in `pass@1` rate compared to the untrained base model. But the model trained with the cross-policy sampling strategy demonstrates a consistent advantage as $k$ increases. This suggests that the strategy successfully preserves the model's diversity while improving its overall ability.

## 5 RELATED WORK

**Reinforcement Learning AI Agents.** RL algorithms like PPO (Schulman et al., 2017) and GRPO (Shao et al., 2024) have been widely adopted in LLM agent training. Deepseek-R1 (DeepSeek-AI et al., 2025) demonstrates RL's ability to incentivize reasoning in LLMs through reward-driven fine-tuning. Recent works (Qian et al., 2025; Feng et al., 2025; Lu et al., 2025a; Wen et al., 2025) further develop RL techniques. GUI agents also benefit from RL-driven optimization (Xu et al., 2024; Qi et al., 2024; Liu et al., 2024b; Qin et al., 2025; Chen et al., 2025). For long-horizon tasks, Chen et al. (2025) shows RL's efficacy in balancing exploration and tool usage. DeepResearcher further scales real-world research by training agents to iteratively refine hypotheses via RL (Zheng et al., 2025b). Despite these advancements, most current approaches fall short in studying the exploration aspect of RL training and the multi-task setting. In this work, we propose the cross-policy sampling strategy and task advantage normalization, addressing a critical gap in existing methods.

**Reinforcement Learning Infrastructure.** Several frameworks (Sheng et al., 2024; Hu et al., 2024; Fu et al., 2025) have been developed for RL training. These frameworks usually adopt modern training (Shoeybi et al., 2019; Zhao et al., 2023) and rollout (Kwon et al., 2023; Zheng et al., 2024) engines to boost efficiency. However, unlike math or coding tasks, agent scenarios involve multi-turn interactions with environments. There have been works (Liu et al., 2024c; Ma et al., 2024) to provide standardized benchmarks for evaluating multi-turn interactions and addressing reproducibility gaps. Platforms such as E2B (e2b dev, 2025) and OpenHands (Wang et al., 2025a) provide secure sandbox environments and modular interfaces for code execution, browser automation, and generalist agent development. While these environments provide strong support for agent evaluation, existing RL frameworks lack built-in support for multi-turn interactions and agent-specific training optimizations.

## 6 CONCLUSION

We propose AGENTRL, a system for training LLM agents with RL across diverse tasks and environments. Through asynchronous rollout–training pipelines, scalable environment deployment, and algorithmic advances including cross-policy sampling and task advantage normalization, AGENTRL enables more efficient and stable training. Experiments demonstrate competitive results across diverse agentic benchmarks, with encouraging signs of generalization to unseen tasks.

STATEMENTS

**Ethics Statement**    This work does not involve human subjects or sensitive personal data. All experiments are conducted on publicly available datasets and environments, and we provide full documentation of preprocessing and implementation details to support transparency. The methods and findings are intended for advancing research on reinforcement learning with LLMs; we do not foresee immediate risks of harmful applications, but we acknowledge the general possibility of misuse of LLM agents. We encourage responsible use of our released resources in line with the ICLR Code of Ethics.

**Reproducibility Statement**    We place a strong emphasis on reproducibility and have made extensive efforts to ensure that our results can be reliably reproduced. To this end, we release all code, environments, and training scripts, together with detailed hyperparameters and configuration files, in our anonymous repository. Additional descriptions of environment setup, data preprocessing, and implementation details are provided in the appendix and supplementary materials. These resources collectively support transparent and reproducible verification of our findings.

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

# A    BACKGROUND OF REINFORCEMENT LEARNING IN LARGE LANGUAGE MODELS

Reinforcement Learning (RL) has significantly enhanced the capabilities of Large Language Models (LLMs) by optimizing their decision-making through reward-driven training. The fundamental RL objective is expressed as:

$$\mathcal{J}(\theta) = \mathbb{E}_{s \sim \mathcal{D}, a \sim \pi_\theta(s)}[R(s,a)], \tag{2}$$

where $\pi_\theta$ denotes the policy, $s$ represents the input context, $a$ is the generated output, and $R(s,a)$ assesses the output quality via a reward function.

**A key method, Proximal Policy Optimization (PPO)**(Schulman et al., 2017), ensures training stability using a clipped probability ratio, defined as:

$$\rho_t(\theta) = \frac{\pi_\theta(a_t|s_t)}{\pi_{\text{old}}(a_t|s_t)}, \tag{3}$$

with the objective function:

$$\mathcal{J}_{\text{PPO}}(\theta) = \mathbb{E}_t[\min(\rho_t(\theta)\hat{A}_t, \text{clip}(\rho_t, 1-\epsilon, 1+\epsilon)\hat{A}_t) - \beta D_{\text{KL}}], \tag{4}$$

where $\hat{A}_t$ is the advantage estimate, and clipping limits policy updates.

For improved advantage estimation, Generalized Advantage Estimation (GAE)(Schulman et al., 2018) is utilized, computed as:

$$\hat{A}_t^{\text{GAE}}(\gamma, \lambda) = \sum_{l=0}^{\infty} (\gamma\lambda)^l \delta_{t+l}, \tag{5}$$

where $\delta_t = r_t + \gamma V(s_{t+1}) - V(s_t)$ is the temporal difference error, and $\gamma$ and $\lambda$ adjust the bias-variance tradeoff.

**Another approach, Group Relative Policy Optimization (GRPO)**(Shao et al., 2024), optimizes over groups of outputs with the objective:

$$\mathcal{J}_{\text{GRPO}}(\theta) = \mathbb{E}_{o_i \sim \pi_{\text{group}}(\theta)}[J_{\text{group}}(\theta)], \tag{6}$$

where the group objective is:

$$\mathcal{J}_{\text{group}}(\theta) = \frac{1}{G}\sum_{i=1}^{G} \min(\rho_i \hat{A}_i, \hat{\rho}_i) - \beta D_{\text{KL}}, \tag{7}$$

and the advantage $\hat{A}_i$ is a normalized reward:

$$\hat{A}_i = \frac{r_i - \mu_r}{\sigma_r}, \tag{8}$$

with $\mu_r$ and $\sigma_r$ as the mean and standard deviation of rewards, fostering adaptive LLM behaviors.

**Finally, Decoupled Clip and Dynamic sampling Policy Optimization (DAPO)**(Yu et al., 2025) was proposed to address issues specific to long-CoT reinforcement learning, such as entropy collapse and training instability. The algorithm modifies the GRPO objective by introducing several key techniques, including a decoupled clipping mechanism and a dynamic sampling strategy.

The DAPO objective function is formulated as:

$$\mathcal{J}_{DAPO}(\theta) = \mathbb{E}_{(q,a)\sim\mathcal{D},\{o_i\}_{i=1}^{G}\sim\pi_{\theta_{old}}(\cdot|q)} \left[ \frac{1}{\sum_{i=1}^{G}|o_i|} \sum_{i=1}^{G}\sum_{t=1}^{|o_i|} \right.$$
$$\left. \times \min\left(r_{i,t}(\theta)\hat{A}_{i,t}, \text{clip}(r_{i,t}(\theta), 1-\epsilon_{low}, 1+\epsilon_{high})\hat{A}_{i,t}\right) \right] \tag{9}$$

subject to the constraint:

$$0 < |\{o_i|\text{is\_equivalent}(a, o_i)\}| < G, \tag{10}$$

where the advantage $\hat{A}_{i,t}$ is calculated similarly to GRPO. The primary innovations are the **decoupled clipping bounds**, $\epsilon_{low}$ and $\epsilon_{high}$, which allow for greater exploration to prevent entropy collapse, and the **dynamic sampling constraint**, which filters out batches where all responses are either correct or incorrect to ensure a non-zero advantage and stable gradients. The loss is also normalized at the **token level** ($\frac{1}{\sum |o_i|}$) to properly weight responses of varying lengths.

# B PRELIMINARIES

## B.1 PROBLEM DEFINITION

**Agentic Task.** We define an *agentic task* $\mathcal{T}_i$ as a Markov Decision Process (MDP):

$$\mathcal{T}_i = \left(\mathcal{S}_i^{\text{env}}, \mathcal{A}_i, P_i, r_i, \rho_i\right), \tag{11}$$

where $\mathcal{S}_i^{\text{env}}$ is the environment state space, $\mathcal{A}_i$ is the action space, $P_i(s'|s, a)$ denotes transition dynamics, $r_i(s, a)$ is the reward function, and $\rho_i(s_0)$ is the initial state distribution.

**LLM-based Policy and Composite State.** When the policy $\pi_\theta$ is implemented by an LLM, the state at decision step $t$ is the *composite state* $s_t = (s_t^{\text{env}}, s_t^{\text{ctx}})$, where $s_t^{\text{env}}$ is the environment state and $s_t^{\text{ctx}} \in \mathcal{V}^*$ is a tokenized context representing the trajectory prefix up to step $t$.

In LLM-based settings, a high-level action $a_t$ is a complete sequence of $L_t$ tokens:

$$a_t = (y_{t,1}, y_{t,2}, \ldots, y_{t,L_t}), \quad y_{t,k} \in \mathcal{V}. \tag{12}$$

The underlying LLM defines a token-level probability distribution $P_\theta(y_{t,k} \mid s_t^{\text{ctx}}, y_{t,<k})$ for each token, and the policy probability of producing $a_t$ from $s_t$ factorizes as:

$$\pi_\theta(a_t \mid s_t) = \prod_{k=1}^{L_t} P_\theta\left(y_{t,k} \mid s_t^{\text{ctx}}, y_{t,<k}\right). \tag{13}$$

This factorization allows us to define token-level log-probabilities and, consequently, token-level policy gradients and advantage estimates.

**Trajectory Definition.** A trajectory in task $\mathcal{T}_i$ is defined as

$$\tau = \left(s^{(0)}, a^{(0)}, r^{(1)}, s^{(1)}, a^{(1)}, r^{(2)}, \ldots, s^{(T-1)}, a^{(T-1)}, r^{(T)}, s^{(T)}\right), \tag{14}$$

where each $s^{(t)} = (s_t^{\text{env}}, s_t^{\text{ctx}})$ is a composite state as above. The reward $r^{(t+1)} = r_i(s_t^{\text{env}}, a^{(t)})$ is assigned after $a^{(t)}$ is applied in $s_t^{\text{env}}$. Different from standard MDP trajectories, this formulation explicitly embeds a context component in each state.

**Multi-Task Setting.** We study a collection of $N_{\text{task}}$ tasks:

$$\mathcal{T} = \{\mathcal{T}_1, \ldots, \mathcal{T}_{N_{\text{task}}}\}. \tag{15}$$

For each $\mathcal{T}_i$ there are $M_i$ samples:

$$\mathcal{D}_i = \{x_{i,1}, \ldots, x_{i,M_i}\}. \tag{16}$$

Executing sample $x_{i,j}$ produces a *group* of $K_{i,j}$ trajectories:

$$G_{i,j} = \{\tau_{i,j,1}, \ldots, \tau_{i,j,K_{i,j}}\}, \tag{17}$$

which, as we will discuss in RLVR, are used in GRPO to compute group-based advantage estimates.

## B.2 REINFORCEMENT LEARNING WITH VERIFIABLE REWARDS (RLVR)

Reinforcement Learning with Verifiable Rewards (RLVR) DeepSeek-AI et al. (2025) refers to scenarios in which the reward signal associated with a trajectory can be computed in a deterministic and objective manner based on the observed interaction data. In practice, RLVR is commonly optimized using Proximal Policy Optimization (PPO) Schulman et al. (2017) or its extensions such as Group Relative Policy Optimization (GRPO) (Shao et al., 2024).

**PPO Objective.** Given a batch of trajectories, PPO maximizes the clipped surrogate objective:

$$\mathcal{L}_{\text{PPO}}(\theta) = \mathbb{E}_t \left[\min\left(r_t(\theta)\hat{A}_t, \, \text{clip}(r_t(\theta), 1 - \epsilon, 1 + \epsilon)\hat{A}_t\right)\right], \tag{18}$$

where $r_t(\theta) = \frac{\pi_\theta(a_t|s_t)}{\pi_{\theta_{\text{old}}}(a_t|s_t)}$ is the probability ratio and $\hat{A}_t$ is the advantage estimate.

**GRPO Objective.** Under GRPO, each group $G_{i,j}$ contains $K_{i,j}$ trajectories compared within the group, yielding group-relative advantage estimates $\hat{A}_{i,j,g}$. The objective is:

$$\mathcal{L}_{\text{GRPO}}(\theta) = \mathbb{E}_{i,j}\left[\frac{1}{K_{i,j}}\sum_{g=1}^{K_{i,j}}\min\left(\rho_{i,j,g}(\theta)\,\hat{A}_{i,j,g},\ \text{clip}(\rho_{i,j,g}(\theta), 1-\epsilon, 1+\epsilon)\,\hat{A}_{i,j,g}\right)\right], \quad (19)$$

where $\rho_{i,j,g}(\theta) = \frac{\pi_\theta(a_{i,j,g}|s_{i,j,g})}{\pi_{\theta_{\text{old}}}(a_{i,j,g}|s_{i,j,g})}$ and $\hat{A}_{i,j,g} = \frac{\hat{R}_{i,j,g}-\text{mean}(\hat{R}_{i,j})}{\text{std}(\hat{R}_{i,j})}$ is a group-relative advantage estimate, computed as the difference between the empirical return $\hat{R}_{i,j,g}$ of trajectory $\tau_{i,j,g}$.

### B.3 FORMAL DESCRIPTION AND THEORETICAL ANALYSIS OF CROSS-POLICY SAMPLING

Formally, let $\mathcal{M} = \{\pi_{\theta_0}, \pi_{\theta_1}, \ldots, \pi_{\theta_K}\}$ denote the set of candidate policies (e.g., the current policy and historical snapshots). Unlike standard sampling where actions are drawn from a single policy, the trajectory obtained by Cross-Policy Sampling (CPS), denoted as $\tau^c$, is generated by dynamically selecting a policy at each step. The trajectory takes the form:

$$\tau^c = \left(s^{(0)}, a^{c,(0)}, r^{(1)}, s^{(1)}, a^{c,(1)}, \ldots, s^{(T)}\right), \quad (20)$$

where at each timestep $t$, the action is sampled via a two-stage process:

$$k \sim \mathcal{U}(0, K), \quad a^{c,(t)} \sim \pi_{\theta_k}(\cdot \mid s^{(t)}). \quad (21)$$

Here, $s^{(t)} = (s_t^{\text{env}}, s_t^{\text{ctx}})$ represents the composite state of the environment and the language context context.

To analyze the exploration benefit, we introduce a geometric interpretation of the language-agent interaction. The language state $s^{\text{ctx}}$ can be viewed as a point in a high-dimensional semantic space $\mathcal{L}$. However, effective communication requires the state to remain within the subspace of linguistically coherent sequences, denoted as $\mathcal{L}_{\text{valid}} \subset \mathcal{L}$. The environment state $s^{\text{env}}$ is determined stochastically by a grounding function $\Gamma : \mathcal{L}_{\text{valid}} \to \Delta(\mathcal{S}^{\text{env}})$.

Let $\mathcal{G} \subset \mathcal{S}^{\text{env}}$ be the set of goal states (success set). We define the *language preimage* of the goal as:

$$\mathcal{L}_{\mathcal{G}} = \Gamma^{-1}(\mathcal{G}) \cap \mathcal{L}_{\text{valid}}. \quad (22)$$

This set $\mathcal{L}_{\mathcal{G}}$ represents all valid thought/action sequences that lead to success. Finding a solution is equivalent to locating a trajectory that intersects with $\mathcal{L}_{\mathcal{G}}$.

The core advantage of CPS lies in its coverage of this preimage. A single policy $\pi_\theta$ tends to collapse to a specific mode (a subset of valid paths). By sampling from a mixture of policies $\pi_{\text{mix}} = \frac{1}{|\mathcal{M}|}\sum_{\pi \in \mathcal{M}} \pi$, CPS effectively computes the union of the support of individual policies. Crucially, since every $\pi \in \mathcal{M}$ is a trained language model, their samples remain confined to the valid manifold $\mathcal{L}_{\text{valid}}$. Thus, the support of the cross-sampled trajectory $\tau^c$ satisfies:

$$\text{supp}(\tau^c) \cap \mathcal{L}_{\mathcal{G}} \approx \bigcup_{\pi \in \mathcal{M}} (\text{supp}(\tau^\pi) \cap \mathcal{L}_{\mathcal{G}}) \supsetneq \text{supp}(\tau^{\text{current}}) \cap \mathcal{L}_{\mathcal{G}}. \quad (23)$$

This inequality highlights that CPS strictly expands the explored region within the valid solution space $\mathcal{L}_{\mathcal{G}}$ compared to the current policy alone.

**Remark on Stability:** It is important to note why this is superior to simply increasing the sampling temperature. High-temperature sampling expands the support isotropically in $\mathcal{L}$, often causing the trajectory to drift into $\mathcal{L} \setminus \mathcal{L}_{\text{valid}}$ (incoherent or hallucinated text). In contrast, CPS expands diversity along the "directions" of previous valid policies, thereby increasing the probability $P(s^{\text{env}} \in \mathcal{G})$ without sacrificing linguistic coherence.

### B.4 MITIGATING THE OFF-POLICY BIAS

The off-policy bias from the asynchronous pipeline is carefully managed at two levels:

**At the algorithmic level:** GRPO, by building upon PPO, inherits Importance Sampling (IS). The probability ratio $r_t(\theta) = \frac{\pi_\theta(a|s)}{\pi_{old}(a|s)}$ corrects for the distributional shift between the current policy ($\pi_\theta$)

and the behavior policy ($\pi_{old}$) that collected the data. To ensure this correction is accurate, the key is obtaining the correct logprobs for $\pi_{old}$. We achieve this by **directly passing the logprobs from the rollout engine along with the trajectory data. This ensures the training objective remains unbiased, even if minor model version discrepancies exist between the training and rollout engines.

**At the pipeline level:** We enforce a strict data flow where all data sent to the data queue at step $N$ serves as the training data for step $N + 1$. This design naturally prevents the data queue from accumulating stale trajectories and ensures the training data is always "as up-to-date as possible."

Furthermore, our investigation revealed that a more significant source of practical off-policy bias stemmed from a subtle tokenization issue: the **token $\rightarrow$ text $\rightarrow$ token mapping is often not identical**. Re-tokenizing multi-turn outputs during training can inadvertently introduce this drift. We have updated the paper to clarify that **AgentRL avoids this effect entirely by preserving the original token sequences** throughout the rollout and training process.

We derive the stale policy from the main policy, which ensures the distributions of the two policies do not diverge significantly. In our setting, the stale engine is synchronized every 25 steps. With the help of the approaches introduced above, this off-policy bias can be implicitly corrected and does not affect training stability.

## C    DATASET DETAILS

### C.1    EXTENDING AGENTBENCH

While the overall framework is decoupled from benchmarks, we perform training on a refined version of AGENTBENCH, or what we call AGENTBENCH-FC. Specifically, we made several modifications:

### C.2    SYNTHESIZING TRAINING SET

To address the scarcity of training data in the original AGENTBENCH framework, we aim to construct a large-scale and diverse dataset suitable for reinforcement learning across various agent environments. To this end, we adopted a multifaceted data collection strategy tailored to the unique characteristics of each environment:

**Direct Adoption of Existing Datasets.** For environments like `AlfWorld` and `WebShop`, which are accompanied by rich, pre-existing training sets, we directly incorporated these official datasets. This approach ensures consistency with the original benchmarks and leverages well-established data sources.

**Synthetic Data Generation via Self-Instruct.** For tasks in more complex environments such as `OS`, `KnowledgeGraph`, and `DB`, where training data is not readily available, we employed the Self-Instruct methodology (Wang et al., 2022). We used high-performance APIs (o3 and claude4-sonnet) to efficiently sample and filter a large volume of high-quality training instances.

**Augmentation with External High-Quality Datasets.** To further enrich the diversity and complexity of our training data, we integrated external, high-quality datasets. Notably, for the `DB` environment, we augmented our dataset with the training samples provided by the BIRD benchmark (Li et al., 2023), a comprehensive text-to-SQL dataset.

### C.3    MODIFICATIONS TO AGENTBENCH ENVIRONMENT

To enhance the flexibility and compatibility of AGENTBENCH, we transformed its five environments into a Function-Call Based framework. We analyzed the distinct action types required by each environment and categorized them accordingly. For each environment, we extracted specific tools following the OpenAI Function Call Format. For instance, in the Knowledge Graph (KG) environment, we identified and implemented seven tools, including `get_relations`, `get_neighbors`, and `count`, among others. Additionally, we modified the interaction logic of each environment to support external requests in the Function Call format, ensuring seamless integration with external systems.

Outlining the refactoring of Controller and Worker interfaces We restructured the interface protocols for the Controller and Worker components in AGENTBENCH to standardize task management and interaction. The `start_sample` interface was introduced to initiate a task, while multi-turn interactions were facilitated through the `interact` interface. To improve Controller oversight, we implemented additional interfaces, such as `list_sessions` and `list_workers`, enabling efficient monitoring of internal worker and session states within the container.

# D   DETAILED EXPERIMENTAL SETTINGS

## D.1   ENVIRONMENTS AND TASKS

We select five representative multi-turn interaction tasks from the AGENTBENCH (Liu et al., 2024c), a comprehensive and evolving benchmark designed to evaluate the reasoning and decision-making capabilities of large language models. These tasks, encompassing operating system interactions, database management, knowledge graph navigation, text-based adventure games, and web shopping scenarios, are chosen for their diverse challenges and ability to assess critical skills such as long-sequence comprehension, contextual tracking, and environmental interaction. The tasks are supported by standardized evaluation protocols and open-source code environments, facilitating robust experimental implementation and framework refinement.

**Unified Reward.** We normalize all task rewards to the range $[0, 1]$ for consistency. For tasks without intrinsic reward signals, we assign a reward of 1 for correct responses and 0 otherwise. In addition, we leverage termination signals and penalize abnormal terminations with a reward of $-0.2$ to encourage proper episode completion.

- **Operating System (OS) Task:** This environment assesses an agent's ability to interact with a real Ubuntu Docker-based operating system through Bash command-line inputs. Agents are tasked with interpreting natural language instructions and translating them into precise Shell commands to achieve specific objectives, such as file manipulation or directory navigation in an unfamiliar environment. The task demands high accuracy in command generation, error handling, and result interpretation (e.g., standard output and error streams), given the vast action space and the need for adaptive decision-making.

- **Database (DB) Task:** In this scenario, agents act as database analysts, interacting with a real database via SQL queries to address natural language questions or perform data modifications (e.g., INSERT, UPDATE). The task evaluates the agent's proficiency in converting natural language to SQL (Text-to-SQL), understanding database schemas (table structures, column names, data types), and managing complex queries (e.g., multi-table joins, nested queries, aggregation functions). Multi-turn interactions require agents to adjust strategies based on query results or error feedback.

- **Knowledge Graph (KG) Task:** For the KG environment, API results are obtained with one-shot testing to ensure the model can correctly invoke tool calls, while our trained models are trained and evaluated without one-shot assistance. This task challenges agents to perform multi-step reasoning and information retrieval within a large knowledge graph (e.g., Freebase) to answer complex queries. With only partial observability due to the graph's scale, agents must use structured query operations (e.g., retrieving entity relationships or finding intersecting entity sets via callable tools) to explore and connect information fragments. It emphasizes long-term planning, information integration, and effective decision-making under incomplete information.

- **Text Adventure Game (Text Game / House-Holding, HH - Represented by ALFWorld):** Agents operate in a text-described virtual household environment, executing action sequences to meet high-level goals (e.g., "clean a soapbar and place it on the workbench"). Actions include navigating (e.g., "go to cabinet 1"), interacting with objects (e.g., "take soapbar 1 from sinkbasin 1"), and adjusting plans based on feedback (e.g., "The cabinet 2 is closed"). ALFWorld (Shridhar et al., 2021) highlights the need for commonsense reasoning, goal decomposition, and dynamic planning in response to environmental states.

- **Web Shopping (WS - Represented by WebShop):** This task simulates an e-commerce experience where agents search for products based on specific criteria (e.g., brand, price) by interacting with a simulated website. Actions include keyword searches, link clicks, attribute filtering, and adding items to a cart. The WebShop environment (Yao et al., 2022) offers a rich product dataset,

requiring agents to analyze requirements, navigate multi-turn interactions, and demonstrate strong information retrieval, comparison, and decision-making skills in a complex web interface.

## D.2 TRAINING AND EVALUATION SETTINGS

**Training.** We leverage the Verl project as a foundation, implementing a fully asynchronous overhaul to develop a novel training framework, AGENTRL, tailored for agentic RL tasks. The framework was applied to train models including Qwen2.5-3B-Instruct, Qwen2.5-7B-Instruct, Qwen2.5-14B-Instruct, Qwen2.5-32B-Instruct, and GLM4-9B. The efficiency of the asynchronous design enabled extensive rollout training across the five selected multi-turn interaction tasks, facilitating large-scale RL with over 1000 steps in a multi-task mixed setting. This prolonged training ensured convergence of model performance across diverse tasks.

The interaction format between models and environments was standardized using the OpenAI Function Call Format. For the Qwen series, RL training commenced directly from the base models. In contrast, the GLM4-9B model required an initial cold-start phase with a limited set of supervised fine-tuning (SFT) data to adapt to the Function Call Format, followed by RL training, ultimately yielding significant performance improvements (see Table 3). Training was conducted on H800 GPUs, with a minimum configuration of 16 GPUs for the 14B model. Scalability was observed, as training efficiency increased with additional GPU resources.

The training process employed the Group Relative Policy Optimization (GRPO) algorithm as the baseline, enhanced with custom modifications (see Section 3.1). Rollouts were performed with a temperature of 0.8, sampling eight times per rollout to ensure diverse action exploration. To maintain consistency across multi-task environments, a binary reward function was designed, assigning a score based on the correctness of the entire trajectory. Trajectories exceeding the maximum interaction rounds or maximum response length incurred a penalty of -0.2. For computational efficiency, SGLang was adopted as the inference engine, paired with the Fully Sharded Data Parallel (FSDP) strategy to optimize RL training.

**Evaluation.** For evaluation, a lightweight eval script was developed using the SGLang engine, seamlessly integrated with the asynchronous framework to enable rapid assessment of task performance. Evaluations were conducted with a temperature of 0.8, averaging results over four consecutive runs per task to ensure reliability. Additionally, a compatible API evaluation script was created to assess model APIs across tasks, supporting endpoints served by vllm or SGLang, with identical parameters (temperature 0.8, four-run average) to maintain consistency.

## D.3 DEPLOYMENT FRAMEWORK DETAILS

As shown in Figure 5, each worker in the new framework operates as a containerized execution unit, capable of managing concurrent task lifecycles under isolated runtime conditions. Workers are equipped with a detailed instrumentation layer for real-time observability, enabling telemetry at both session and task granularity. Internally, each worker integrates an abstract environment controller that mediates between task definitions and environment provisioning services. This controller is responsible for session instantiation, interaction handling, timeout enforcement, and environment cleanup. By abstracting the execution logic from physical deployment details, the worker layer can accommodate diverse backend configurations and support dynamic elasticity under shifting training loads.

The new controller adopts a non-blocking dispatch strategy that minimizes contention and ensures deadlock safety through a strict lock acquisition hierarchy. Timeout-driven fault detection and self-healing routines enable automatic de-registration and reintegration of unstable nodes. The controller also enforces strict lifecycle policies on session expiration, interaction timeout, and stale data cleanup through periodic maintenance loops.

## D.4 RESULTS ANALYSIS

We provide a comprehensive evaluation of reinforcement learning (RL) performance across a diverse set of models and tasks. We report results for prominent API-based models and popular open-source base models. Additionally, we assess the RL-enhanced variants of our models at various scales,

trained using the AgentRL framework. The evaluation extends to out-of-distribution (OOD) testing on an unseen benchmark, where the RL-trained model demonstrates performance gains over its base counterpart. Furthermore, we conduct an ablation study to investigate the impact of our proposed algorithmic techniques on model efficacy.

**Scaling Law**    The main results reflect a clear scaling law trend, with AgentRL-trained models showing consistent performance improvements as their size increases. Performance progressively escalates from the smallest model variants to the largest, indicating the framework's scalability and robustness. This progressive enhancement underscores the algorithm's adaptability to varying model sizes. The successful application to a model from a different architectural family further validates the framework's versatility, demonstrating its broad applicability beyond a single model series.

**Frontier Model Performance**    Comparative analysis highlights the superiority of our largest AgentRL-trained model over leading API-based models. While prominent proprietary LLMs achieve high scores, our RL-optimized model reaches a new state-of-the-art performance level, representing a substantial improvement over its base version before RL training. This suggests that AgentRL not only competes with but, in certain multi-turn and overall metrics, surpasses these advanced models, affirming its competitive edge.

**Open-Source Baselines**    To rigorously benchmark our framework, we compare against a diverse set of representative open-source methods covering supervised fine-tuning (SFT), pre-training, and evolutionary paradigms. **AgentLM** (Zeng et al., 2024) and **AgentFlan** (Chen et al., 2024a) represent the SFT paradigm; AgentLM employs hybrid instruction tuning on expert trajectories, while AgentFlan focuses on decomposing and cleaning data distributions to mitigate hallucinations. **Hephaestus**(Zhuang et al., 2025) adopts a continual pre-training paradigm, utilizing large-scale agentic corpora to enhance fundamental capabilities like tool understanding before fine-tuning. Finally, **AgentGym**(Xi et al., 2025) serves as a framework baseline that facilitates agent self-evolution through interactive environments. In contrast to these approaches, which primarily focus on static data engineering or synchronous iteration, AGENTRL introduces a fully asynchronous *online* reinforcement learning framework, specifically optimized for multi-turn, multi-task stability via our proposed Cross-Policy Sampling and Task Advantage Normalization.

**OOD Performance**    The OOD evaluation on the BFCL-v3 benchmark tests generalization on unseen tasks. The RL-trained model shows a clear improvement in overall performance compared to the base model, with a particularly significant leap in multi-turn task capability. This outperformance after extensive RL training underscores the method's ability to generalize beyond its training distribution, enhancing its potential for practical deployment in diverse scenarios.

**Ablation Study**    The ablation study further elucidates the efficacy of our methodological enhancements, detailed as follows:

- **Cross-Policy Sampling**: This technique, designed to explore more states in open-ended environments, proves to be highly effective. Its inclusion boosts the average performance significantly. This result underscores the value of encouraging broader exploration, as the strategy successfully expands the model's capability boundaries by exposing it to more diverse and goal-relevant trajectories during training.
- **Task Advantage Normalization**: In contrast, this method stabilizes multi-task learning by mitigating negative interference and rate disparities across tasks. These findings support the selective integration of this technique, enhancing AgentRL's training stability and consistency.

## D.5    ADDITIONAL EXPERIMENTS AND ABLATION STUDIES

In this section, we present detailed experimental results to address reviewer inquiries regarding algorithmic effectiveness and system robustness. **Note on Experimental Settings:** To isolate the specific effects of Cross-Policy Sampling and the Asynchronous Pipeline, we conducted targeted experiments on the **DB environment**. For system-wide stability analyses (Task Advantage Normalization and Hyperparameter Sensitivity), we utilized the full **multi-task setting across all five AgentBench environments**.

### D.5.1 ANALYSIS OF ASYNCHRONOUS PIPELINE

**Synchronous vs. Asynchronous Pipeline (Single Task - DB).** To address concerns regarding potential off-policy bias, we conducted a comparative experiment between synchronous and asynchronous pipelines on the DB environment. As shown in Figure 9, the training curves of the two approaches are nearly identical. This empirical evidence confirms that the off-policy bias introduced by the asynchronous mechanism has a negligible impact on convergence and performance, while retaining the substantial efficiency gains demonstrated in the main paper.

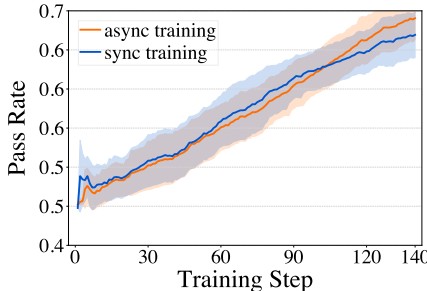

Figure 9: Async vs. Sync (**DB Environment**). Asynchronous training yields nearly identical convergence, confirming negligible off-policy bias.

### D.5.2 HYPERPARAMETER SENSITIVITY ANALYSIS

**Algorithm Sensitivity (Multi-Task).** Figure 10 illustrates the training trajectories under different hyperparameter settings across the full multi-task suite. While we could only conduct limited additional sweeps due to resource constraints, the results demonstrate that with the aid of our proposed algorithmic components, the framework exhibits strong tolerance to hyperparameter adjustments. The model maintains a competitive trajectory even when parameters deviate from the local optimum.

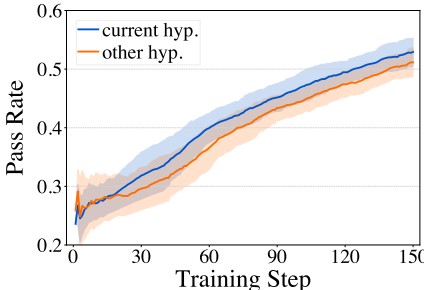

Figure 10: Hyperparameter Sensitivity (**Multi-task**). The framework exhibits tolerance to hyperparameter adjustments in multi-task settings.

### D.5.3 EFFECTIVENESS OF ALGORITHMIC COMPONENTS

We investigate the impact of our two proposed algorithmic improvements under their respective validation settings.

**Effect of Cross-Policy Sampling (Single Task - DB).** To verify the motivation and effectiveness of Cross-Policy Sampling, we conducted a supplementary comparison on the single-task DB environment. As shown in Figure 11(a), the difference between the two settings is significant. With Cross-Policy Sampling enabled (blue line), we observe a distinct performance surge, whereas the curve without it (orange line) remains consistently at a lower level. This confirms that Cross-Policy Sampling is essential for sustaining exploration in complex single-task scenarios.

**Effect of Task Advantage Normalization (Multi-Task - Sub-optimal Hyp.).** To demonstrate the practical utility of Task Advantage Normalization (TAN), we conducted a specific experiment across the five-task suite using a *sub-optimal* set of hyperparameters. As shown in Figure 11(b), Task Advantage Normalization plays a critical role in stabilizing training and boosting performance under these conditions. While marginal improvements might appear less pronounced under the carefully tuned hyperparameters used in the main paper, this experiment highlights Task Advantage Normalization's value in ensuring robustness when optimal hyperparameters are unknown.

### D.5.4 COMPARISON WITH EXPERIENCE REPLAY

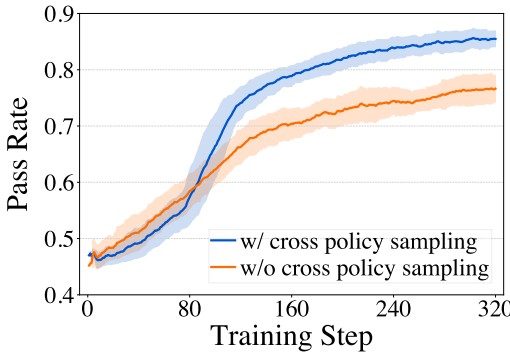
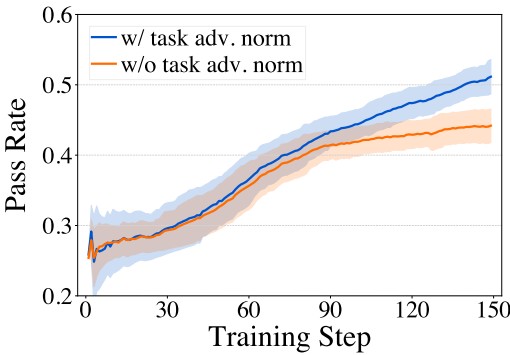

(a) Effect of Cross-Policy Sampling (**DB Environment**)

(b) Effect of Task Adv. Normalization (**Multi-task**)

Figure 11: Ablation study on core algorithmic designs. (a) Cross-Policy Sampling significantly boosts performance on the **DB task**. (b) Task Advantage Normalization stabilizes training across **heterogeneous multi-task settings**, especially under sub-optimal hyperparameters.

We evaluate the efficacy of Cross-Policy Sampling (CPS) by comparing it against a standard Experience Replay (ER) baseline on the **DB environment**. This comparison aims to isolate the benefits of dynamic policy mixing versus replaying historical trajectories.

As illustrated in Figure 12, the two methods demonstrate divergent training dynamics. **Experience Replay (Orange Line)** exhibits superior sample efficiency in the initial phase (steps 0–80), benefiting from the reuse of previous transitions. However, performance rapidly saturates at a pass rate of approximately 79%. We attribute this premature plateau to the *static* nature of replayed trajectories; in multi-turn agentic tasks, as the current policy updates, the distribution mismatch between the behavioral policy (from the replay buffer) and the target policy grows, leading to high-variance importance sampling weights and detrimental off-policy bias.

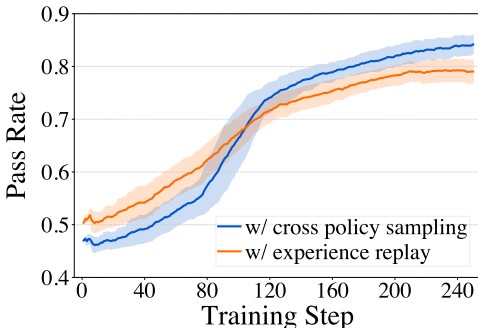

Figure 12: Comparison on DB task. **ER (orange)** plateaus early due to static off-policy bias, while **CPS (blue)** sustains exploration and achieves a significantly higher final success rate.

In contrast, **Cross-Policy Sampling (Blue Line)** shows a steady and sustained improvement trajectory. Although the initial growth is slower due to the exploration variance introduced by mixing policies, CPS overtakes ER around step 100 and achieves a significantly higher asymptotic performance (∼84%). This suggests that CPS effectively mitigates off-policy issues by mixing policies during the *generation* phase, thereby ensuring that the explored trajectories remain linguistically coherent and on-manifold while effectively expanding the state space coverage.

### D.6 CASE STUDIES

#### D.6.1 CASE STUDY ON THE EFFICACY OF CROSS-SAMPLING

To intuitively demonstrate the effectiveness of our proposed cross sampling strategy, we present a case study on a specific knowledge graph (KG) question-answering task. As shown in fig 13, we analyze the execution trajectories of two models, **GLM-4-9B** and **Llama-8B**, on this task. The results show that when tasked individually, both models fail for different reasons. However, when applying our Cross-Policy Sampling strategy, the agent successfully completes the task by finding the correct answer.

The failures of the two individual models stem from distinct causes. **GLM-4** becomes trapped in a **premature conclusion loop**; it correctly deduces the final answer through logical inference but consequently bypasses the required protocol of using tools for verification. It repeatedly outputs its

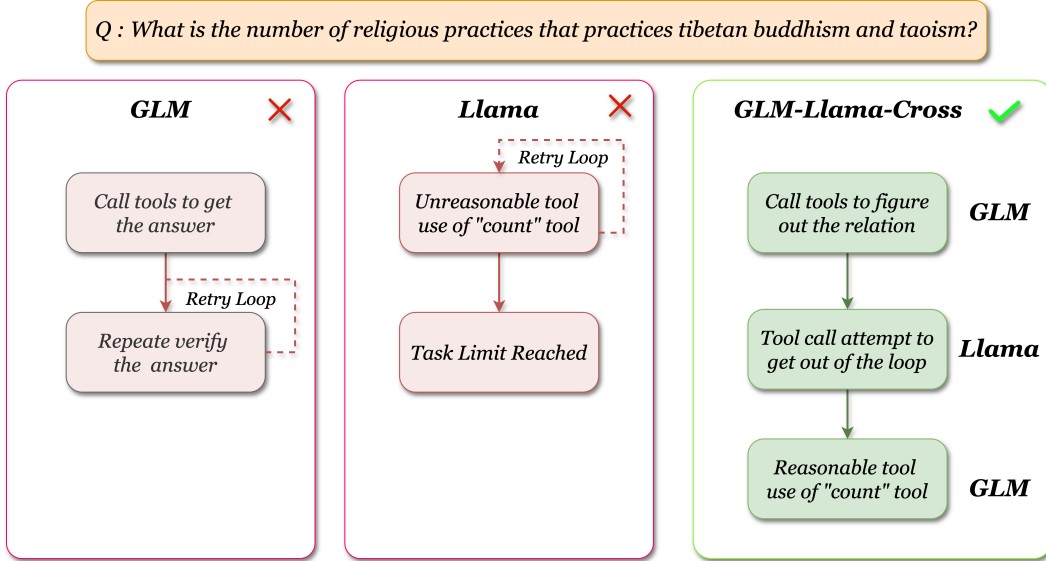

Figure 13: An example of GLM,Llama and GLM-Llama cross sampling in a KG task. This case study demonstrates the Cross-Policy Sampling strategy's success on a KG question-answering task, where GLM-4 fails in a conclusion loop and Llama falters with tool comprehension. It combines GLM-4's logic with Llama's tool interaction to achieve the correct answer.

inferred conclusion in a non-standard format, leading to failure. In contrast, **Llama**'s failure is due to **flawed tool comprehension**; it persistently attempts to call tools with incorrect logic and parameters, indicating a fundamental misunderstanding of the tools' functionality and usage, which prevents any effective progress on the task.

The cross sampling strategy's success stems from a synergy that compensates for each model's weaknesses. It leverages GLM-4's strong logical planning to set a course, then breaks GLM-4's resulting non-interactive loop by switching to Llama's policy. Although Llama's own tool comprehension is flawed, its policy's critical function is to force an attempt at tool interaction. This switch to a "tool-centric" mode, guided by GLM-4's original logic, creates the opportunity for a valid tool call to emerge. This case study highlights the superiority of Cross-Policy sampling by showing how it dynamically combines different problem-solving approaches to forge a successful path where single agents fail.

### D.6.2 PERFORMANCE SCALING WITH MORE THAN TWO POLICIES

**Performance Scaling with Multi-Model Cross-Policy Sampling.** To further validate the scaling potential of Cross-Policy Sampling (CPS) beyond intra-family checkpoints, we conducted an inference experiment on the **DB environment** using three distinct models: Qwen2.5-14B, Qwen3-14B, and GLM-4-9B. As detailed in Table 7, while the individual models exhibit varying performance levels (with GLM-4 struggling on this specific task), the mixed Cross-Policy strategy successfully aggregates their capabilities. Notably, at *pass@64*, the Cross-Policy approach achieves a success rate of **75.7%**, surpassing the best single model (Qwen2.5-14B at 74.0

**Scaling Analysis on WebShop.** We extended the scaling analysis to the **WebShop environment**. As presented in Table 8, we integrated Qwen2.5-14B, Qwen3-14B, and GLM-4-9B. This experiment reveals an interesting trade-off between average quality and diversity. At low sample counts ($k \leq 16$), the inclusion of the weaker GLM-4 model (5.3% at pass@1) dilutes the ensemble's precision, resulting in performance slightly below the best single model. However, as $k$ increases, the benefit of diversity becomes dominant. The Cross-Policy Sampling strategy successfully overtakes the strongest single model (Qwen2.5-14B) at pass@32 and achieves **58.7%** at pass@64 (vs. 56.8%). This confirms that even when constituent policies have mixed quality, the ensemble effectively expands the search coverage to uncover solutions that single policies miss.

Table 7: Pass@$k$ performance comparison on the **DB Environment** using distinct models and their Cross-Policy combination. The Cross-Policy Sampling strategy (mixing Qwen2.5, Qwen3, and GLM-4) achieves the highest coverage at larger $k$, demonstrating the benefit of policy diversity.

| Model | P@1 | P@2 | P@4 | P@8 | P@16 | P@32 | P@64 |
|---|---|---|---|---|---|---|---|
| GLM-4-9B | 3.2 | 5.6 | 9.6 | 15.5 | 22.6 | 29.6 | 36.1 |
| Qwen3-14B | **53.9** | **60.3** | 64.3 | 67.4 | 70.0 | 72.0 | 73.5 |
| Qwen2.5-14B | 48.6 | 58.7 | 64.1 | 67.0 | 69.3 | 71.6 | 74.0 |
| **Cross-Policy Sampling** | 49.6 | 59.6 | **65.5** | **69.0** | **71.5** | **73.7** | **75.7** |

Table 8: Pass@$k$ performance comparison on the **WebShop Environment**. At lower $k$, the inclusion of a weaker model (GLM-4) impacts precision. However, at higher $k$ ($k \geq 32$), the diversity gain from Cross-Policy Sampling allows it to outperform the best single model, demonstrating superior solution coverage.

| Model | P@1 | P@2 | P@4 | P@8 | P@16 | P@32 | P@64 |
|---|---|---|---|---|---|---|---|
| GLM-4-9B | 5.3 | 8.2 | 11.4 | 14.6 | 17.7 | 20.9 | 24.5 |
| Qwen3-14B | **16.0** | 22.0 | 28.6 | 35.5 | 41.8 | 46.9 | 50.8 |
| Qwen2.5-14B | 15.6 | **23.1** | **30.6** | **38.0** | **45.1** | 51.5 | 56.7 |
| **Cross-Policy Sampling** | 15.5 | 22.2 | 29.7 | 36.7 | 43.2 | **52.1** | **58.7** |

### D.6.3 SENSITIVITY TO POLICY MIXING RATIO.

**Sensitivity to Policy Mixing Ratio (WebShop).** We investigated mixing ratio sensitivity on **WebShop** using Qwen3-14B and Qwen2.5-14B. As shown in Table 9, while all cross-policy combinations outperform single models at high $k$, the Ratio 1:2 (favoring Qwen2.5) achieves the highest ceiling (**62.5%** at pass@64). This indicates that maximizing solution coverage requires a strategic balance: injecting sufficient diversity from the auxiliary policy (Qwen3) while maintaining a higher sampling probability for the model with superior intrinsic search capabilities (Qwen2.5).

Table 9: Sensitivity analysis of mixing ratios (Qwen3-14B : Qwen2.5-14B) on the **WebShop task**. While all mixing strategies surpass single models at high $k$, favoring the stronger model (Ratio 1:2) yields the highest performance ceiling (62.5% at P@64), demonstrating the optimal trade-off between diversity and model capability.

| Model / Ratio (Q3:Q2.5) | P@1 | P@2 | P@4 | P@8 | P@16 | P@32 | P@64 |
|---|---|---|---|---|---|---|---|
| Qwen3-14B (Single) | 16.0 | 22.0 | 28.6 | 35.5 | 41.8 | 46.9 | 50.8 |
| Qwen2.5-14B (Single) | 15.6 | 23.0 | 30.6 | 38.0 | 45.1 | 51.5 | 56.7 |
| Ratio 2:1 (Favor Q3) | 16.6 | 23.3 | 30.7 | 38.0 | 44.6 | 50.9 | 57.5 |
| Ratio 1:1 (Balanced) | 16.7 | 23.7 | 31.6 | 39.6 | 47.0 | 53.9 | 60.0 |
| **Ratio 1:2 (Favor Q2.5)** | **16.8** | **24.0** | **31.8** | **39.9** | **47.9** | **55.4** | **62.5** |

### D.6.4 ERROR ANALYSIS

We analyze the performance of the Qwen2.5-14B-Instruct model and the AGENTRL model across five environments (AlfWorld, DB, KG, OS, WebShop), focusing on the primary termination states: Completed and Task Limit Reached.

The data highlights a substantial improvement with the AGENTRL method, where Completed rates increase significantly (e.g., from 0.070 to 0.926 in AlfWorld) and Task Limit Reached rates decrease

| Environment | Base Model | | AGENTRL Model | |
|---|---|---|---|---|
| | Completed | Task Limit Reached | Completed | Task Limit Reached |
| AlfWorld | 0.070 | 0.68 | 0.926 | 0.074 |
| DB | 0.957 | 0.043 | 0.993 | 0.007 |
| KG | 0.747 | 0.213 | 0.947 | 0.033 |
| OS | 0.548 | 0.444 | 0.847 | 0.118 |
| WebShop | 0.725 | 0.275 | 0.980 | 0.020 |

Table 10: Failure Modes Comparison. *Note: "Completed" indicates the agent submitted an answer, not necessarily correctly, and these two statuses are not exhaustive; the sum of percentages may not reach 100% due to other possible outcomes.*

(e.g., from 0.68 to 0.074 in AlfWorld). This suggests that RL training enhances the model's efficiency, reducing instances where tasks terminate due to time constraints and boosting successful completions across all environments.

### D.6.5 WHAT REINFORCEMENT LEARNING TEACHES MODELS IN ALFWORLD

We analyze a task from ALFWorld where the agent must place a saltshaker in a drawer. We compare the base model (Qwen2.5-14B-Instruct), which fails in four runs, with the RL-trained model (AgentRL-Qwen2.5-14B-Instruct), which succeeds in all four, to highlight RL's impact.

**Base Model Performance**   The base model struggles with:

- **Improper Tool Usage**: Repeatedly attempts invalid actions (e.g., `look`) without using the `take_action` tool, leading to errors.
- **Ineffective Strategy**: Fixates on cabinets (e.g., `cabinet 1`) without exploring likely locations like countertops, resulting in failure.

**RL-Trained Model Performance**   The RL-trained model excels by:

- **Correct Tool Usage**: Consistently uses `take_action` correctly, avoiding procedural errors.
- **Efficient Search**: Prioritizes countertops, quickly finding the saltshaker on `countertop 3`.
- **Action Sequencing**: Navigates to `drawer 1`, opens it, and places the saltshaker, completing the task.

From the above analysis we can see that reinforcement learning significantly enhances the model's performance in ALFWorld by imparting tool proficiency for correct use of environment tools, strategic exploration to prioritize likely locations, and effective action planning for sequencing tasks, enabling efficient, goal-directed behavior that starkly contrasts with the base model's repetitive failures.

### D.7 COMPUTATION COSTS

We report the computational resources required for our main experiments to demonstrate the scalability of the AgentRL framework. All training sessions were conducted on a compute cluster consisting of 4 nodes, with each node equipped with 8 GPUs (totaling 32 GPUs). The specific hardware utilized offers a peak performance of approximately 1500 TFLOPs (BF16) per device.

Table 11 summarizes the computational costs for the 14B and 32B model experiments. Both models were trained for 1,500 steps. Notably, the reported figures account for the full training lifecycle, explicitly including the computational overhead incurred by system interruptions and training resumption. The ability to complete 32B-parameter model training in about 101 hours on 32 GPUs demonstrates the high throughput and efficiency of our asynchronous pipeline.

Table 11: Computational cost breakdown for main experiments. The training was conducted on a cluster of 32 GPUs (4 nodes $\times$ 8 GPUs). GPU hours include resumption overhead.

| Model Size | Training Steps | Est. GPU Hours | Wall-Clock Time | Throughput Efficiency |
|---|---|---|---|---|
| AgentRL-14B | 1,500 | $\sim$1,888 | $\sim$59 hours | 32 GPUs |
| AgentRL-32B | 1,500 | $\sim$3,232 | $\sim$101 hours | 32 GPUs |

# E  PROMPT EXAMPLES

## E.1  ALFWROLD TASK

## E.2  KNOWLEDGE GRAPH (KG) TASK

1512
1513
1514
1515
1516
1517
1518
1519
1520
1521
1522
1523
1524
1525
1526
1527
1528
1529
1530
1531
1532

**System Prompt for AlfWorld**

Interact with a household to solve a task. Imagine you are an intelligent agent in a household environment and your target is to perform actions to complete) the task goal.

At the beginning of your interactions, you will be given the detailed description of the current environment and your goal to accomplish. A tool will be provided for you to use to submit the action you want to take. This tool is the only tool you should and must take in order to operate any action in the environment. The way you perform action is to place the action chosen by you in the arguments field of your tool call.

For each of your turn, you will be given a list of actions which you can choose one to perform in this turn. The action you would like to take should be offered in this format: the name of your next action, and you should fill it in the argument field of your tool call. Note that you should always call a tool to operate an action from the given choices. After your each turn, the environment will give you immediate feedback based on which you plan your next few steps. if the environment output Nothing happened, that means the previous action is invalid and you should try more options.

**Reminder:**

- the action must be chosen from the given available actions. Any actions except provided available actions will be regarded as illegal.
- Always call the tool to hand in your next action and think when necessary.

1533
1534
1535
1536
1537
1538
1539
1540
1541
1542

**System Prompt for Knowledge Graph**

**Instructions:** You are an intelligent agent tasked with answering questions based on the knowledge stored in a **knowledge base (KB)**. Utilize the provided tools to probe the KB and retrieve relevant information to address user queries effectively.

Navigate the KB to identify **relationships**, **attributes**, and **intersections**. where applicable, ensuring the most pertinent information is used to formulate answers.

**Remember:**

- A variable can be an entity or a set of entities resulting from previous queries.
- Ensure the tool selected aligns with the question's demands, following a logical order (e.g., fetch relations before finding neighbors).
- After generating a variable, assess whether it constitutes the **final answer**. Variables are assigned IDs starting from 0 (e.g., #0, #1, etc.).
- Upon identifying the **final answer**, respond with 'Final Answer: #id', where #id is the variable's ID (e.g., 'Final Answer: #3'). Do not invoke tools after determining the final answer!
- Execute one action at a time, with a maximum of 15 actions to find the answer.
- Use the supplied tools unless the **final answer** is identified.

Your thoughtful application of these tools and careful consideration of interactions will guide you to correct answers. Note that the task must be completed within 15 rounds— plan your attempts accordingly!

1543
1544
1545
1546
1547
1548
1549
1550
1551
1552
1553
1554
1555
1556
1557
1558
1559
1560
1561
1562
1563
1564
1565

### E.3 DB Task

---

**System Prompt for DataBase**

I will ask you a question, then you should help me operate a **MySQL database** with SQL to answer the question.You have to explain the problem and your solution to me and write down your thoughts.After thinking and explaining thoroughly, every round you can choose to **operate or to answer** with the two specific tools provided.

If you should execute a SQL query, use the 'execute_sql' function, Your SQL should be in one line. Every time you can only execute one SQL statement. I will only execute the statement in the first SQL code block. Every time you write a SQL, I will execute it for you and give you the output. If you are done operating, and you want to commit your final answer, then use the 'commit_final_answer' function. DO NOT use this tool unless you are sure about your answer. I expect an accurate and correct answer.Your answer should be accurate. Your answer must be exactly the same as the correct answer.If the question is about modifying the database, then after done operation, your answer field can be anything.If your response cannot match any pattern I mentioned earlier, you will be judged as FAIL immediately.You should always use the tools provided to submit your answer. Be careful not to write it in the content field.Your input will be raw MySQL response, you have to deal with it by yourself.

---

### E.4 OS Task

---

**System Prompt for Operating System**

You are an assistant that will act like a person. I will play the role of a **Linux (Ubuntu) operating system**. Your goal is to implement the operations required by me or answer the questions proposed by me.

For each of your turns, you should first think about what you should do, and then call exactly one of the provided tools according to the situation.If you think the output is too long, I will truncate it. The truncated output is not complete. You have to deal with the truncating problem by yourself.

**Attention**, your bash code should not contain any input operation. Once again, you should use one tool in each turn, and should not respond without function calling.

Note that if you think the task has been finished, or there is some message missing to completely complete the task, you should respond with calling the function f̈inish_action, as no additional information will be provided.

Also, note that if you have gotten the answer to the question, you should call the änswer_actiontool instead of simply writing your answer in your response.

Your answers should be exact and precise (for example, a single number), do not answer with full sentences or phrases.Always use a tool provided instead of simply responding with content.

---

### E.5 Webshop Task

---

**System Prompt for Web Shopping**

You are web shopping. I will provide **instructions** about what to do, and you must follow them strictly. Every round, you will receive an observation and a list of available actions. You must respond by calling a tool based on the current state and instructions.

- You can use the **search tool** if it is available.
- You can click one of the buttons in **clickables**.
- If an action is not valid, perform nothing.

Keywords for the **search tool** are your choice, but the value for a click must be from the list of available actions. Remember to design search keywords carefully.

First, think about what to do, then call a tool accordingly. You should always use a tool, even if you have questions to confirm, and you can use any available tool without user permission.

---

# F  DISCUSSIONS

## F.1  LIMITATIONS

While our framework establishes a new state-of-the-art in agentic RL, we identify two primary areas for future research that build upon our solid foundation. First, our novel cross-policy sampling strategy is a key driver of enhanced exploration. By its very design of integrating diverse policies, it can introduce minor distributional shifts. These shifts can manifest as mild, transient instabilities in training dynamics, a manageable trade-off for achieving broader state-space coverage. Future work could explore principled refinements, such as adaptive policy weighting, to further optimize this powerful mechanism. Second, as a foundational work, this paper focuses on rigorously validating our framework across a comprehensive suite of controlled environments. Having established the system's robustness and scalability, the natural next step is its application to more complex and dynamic real-world scenarios. We believe our framework provides the ideal testbed for tackling this exciting challenge.

## F.2  FUTURE WORKS

Looking ahead, we plan to extend AGENTRL to a broader range of environments and scale it to larger models. Future research will also explore more sophisticated variants of cross-policy sampling and develop improved methods for multi-task optimization. We believe these are crucial steps toward creating more general and capable LLM agents.

# G  USE OF LLMS

During the preparation of this manuscript, we used large language models (LLMs) to assist with language polishing and grammar improvement. All research ideas, methods, experiments, and analyses were conceived, designed, and validated by the authors.