# OpenReview forum: "AgentRL: Scaling Agentic Reinforcement Learning with a Multi-Turn, Multi-Task Framework"
_ICLR.cc/2026/Conference — Submitted to ICLR 2026_

### Official Review · Reviewer_3aoe · 2025-10-18

**Soundness:** 3
**Presentation:** 3
**Contribution:** 3
**Rating:** 6
**Confidence:** 3

**Summary:**

This work presents the AGENTRL framework for scalable multi-turn, multi-task agentic RL training. On the infrastructure side, AGENTRL
features a fully-asynchronous generation-training pipeline for efficient multi-turn RL. To support heterogeneous environment development in multi-task RL, authors design a unified function-call based API interface, containerized environment development, and a centralized controller. On the algorithm side, authors propose cross-policy sampling to encourage model exploration in multi-turn settings and task advantage normalization to stabilize multi-task training.

**Strengths:**

1, Develop an asynchronous, multi-task framework AGENTRL for scalable agentic RL training and robust heterogeneous environment deployment.
2,  Design a cross-policy sampling strategy to encourage exploration in multi-turn settings and task advantage normalization to stabilize multi-task RL training.
3, The experiments are comprehensive, including different models on different tasks.
4, The github seems good.

**Weaknesses:**

Please see questions for the details.

**Questions:**

So I am not very familiar with this topic. Could you maybe further explain why you choose Hephaestus and AgentLM as your baseline specifically and what's the difference between your proposed method and those two methods?

---

> ### Author Response · Authors · 2025-11-23
>
> We sincerely thank the reviewer 3aoe for the positive assessment and recognizing our framework's scalability and algorithmic contributions.
>
> > Q1 : So I am not very familiar with this topic. Could you maybe further explain why you choose Hephaestus and AgentLM as your baseline specifically and what's the difference between your proposed method and those two methods?
>
> ### What's the Difference
>
> A fundamental difference is that we use Reinforcement Learning (RL) and they mainly use Supervised Fine-Tuning (SFT).
>
>
> To be more specific, Hephaestus adopts a continual pre-training paradigm to enhance fundamental capabilities via large-scale corpora, while AgentLM utilizes a hybrid instruction tuning approach that performs SFT on expert trajectories. In contrast, AgentRL advances beyond these static learning paradigms by introducing an online RL framework. Specifically，AgentRL introduces:
> - **An efficient agentic RL infrastructure**: including a fully asynchronous, multi‑turn rollout–training pipeline and a unified function‑call API with containerized, centrally orchestrated environments that scale heterogeneous tasks;
> - **Effective learning algorithms for multi-turn & multi-task agentic RL**: cross‑policy sampling to preserve exploration in multi‑turn settings and task advantage normalization to stabilize multi‑task optimization across varied sequence lengths and reward scales.
>
> ### Why choosing them
>
> We choose Hephaestus and AgentLM because they are representative, publicly documented agent training methods on AgentBench-like tasks with comparable setups; they combine instruction-tuned backbones with agent‑specific training and tool use, making them strong and widely cited baselines.
>
> To ensure a comprehensive comparison, we will incorporate these detailed clarifications into the final manuscript. Additionally, we have identified other emerging agent training methods, such as Agent-FLAN, and will include them in the revised version to further enrich our baseline landscape. Finally, we have fully open-sourced our code and configurations to actively contribute to the research community.

---

> > ### Comment · Reviewer_3aoe · 2025-11-25
> >
> > Thanks for your response. I increased my score to 8 for your hard working. Good luck!

---

> > > ### Author Response · Authors · 2025-11-26
> > >
> > > We sincerely thank you for your encouraging response and strong support! We are deeply grateful for your recognition of our efforts to improve the manuscript.

---

### Official Review · Reviewer_rknx · 2025-10-30

**Soundness:** 2
**Presentation:** 3
**Contribution:** 2
**Rating:** 6
**Confidence:** 3

**Summary:**

AGENTRL introduces a scalable, asynchronous multi-task RL framework for training LLM agents in multi-turn interactive environments. It combines cross-policy sampling to boost exploration and task-advantage normalization to stabilize training across heterogeneous tasks. Trained on five agent benchmarks, a single AGENTRL model matches specialist agents and outperforms GPT-5, Claude-Sonnet-4 and DeepSeek-R1, showing strong generalization to unseen tasks.

**Strengths:**

- The ablation studies are detailed and show the individual and combined effects of the two main algorithmic improvements.
- The introduction of cross-policy sampling to enhance exploration and task advantage normalization to address heterogeneity in multi-task RL are practical and well-motivated improvements.

**Weaknesses:**

- Mixing trajectories from different policy checkpoints introduces measurable off-policy bias, maybe more experiments and analysis to describe this bias is needed.
- Missing hyper-parameter sensitivity analysis: Kcross-policy sampling probability, advantage-normalisation batch size, async queue capacity are fixed to single values without grid-search or Sobol sweeps, so robustness of the reported gains is unclear.

**Questions:**

- Can the authors provide more granular per-task and per-instance breakdowns of reward scaling and failure modes to better demonstrate the stability of task advantage normalization?
- How sensitive is AGENTRL to choice of reward normalization and environment design? Is there any failure cases if task heterogeneity is heightened further?

---

> ### Author Response · Authors · 2025-11-23
>
> We sincerely thank the reviewer rnkx for their positive review and constructive comments. We provide our feedback as follows.
>
> ## Cross Policy Sampling
>
> > W1 : Mixing trajectories from different policy checkpoints introduces measurable off-policy bias, maybe more experiments and analysis to describe this bias is needed.
>
>
>
>
> ### Analysis
>
> This measurable bias is effectively neutralized by the **importance sampling mechanism within GRPO**, combined with our specific **engineering mechanisms and design choices**. These measures ensure that the bias remains controlled and does not compromise training stability. We have added this analysis to the main paper.
>
> ### Experiment
>
> To further validate this point, we conducted a comparative experiment with and without cross-policy sampling specifically on the single-task DB environment.
>
> | Method / Training pass rate | Step 40 | Step 80 | Step 120 | Step 160 | Step 200 | Step 240 | Step 280 | Step 320 |
> |---|---|---|---|---|---|---|---|---|
> | w/ cross policy sampling | 0.49 | 0.57 | 0.74 | 0.79 | 0.82 | 0.84 | 0.85 | 0.85 |
> | w/o cross policy sampling | 0.51 | 0.58 | 0.66 | 0.70 | 0.73 | 0.74 | 0.76 | 0.77 |
>
> The model with cross-policy sampling enabled exhibits a distinct performance surge, continuously improving throughout training, while the baseline without CPS stagnates at a lower performance level (also shown in Figure 11a, Appendix D.5.3). **This empirically confirms that cross-policy sampling is essential for preventing suboptimal convergence and does not affect training stability.**
>
>
> ## Hyperparameter
>
> > W2 : Missing hyper-parameter sensitivity analysis: Kcross-policy sampling probability, advantage-normalisation batch size, async queue capacity are fixed to single values without grid-search or Sobol sweeps, so robustness of the reported gains is unclear.
>
>
>
> ### Hyperparameter in Cross-Policy Sampling
>
> To address this, we tested sampling probabilities $\rho \in \{1/3, 1/2, 2/3\}$ on the WebShop environment.
>
>
> | Model / Ratio (Q3:Q2.5) | P@1 | P@2 | P@4 | P@8 | P@16 | P@32 | P@64 |
> | :--- | :---: | :---: | :---: | :---: | :---: | :---: | :---: |
> | Qwen3-14B (Single) | 16.0 | 22.0 | 28.6 | 35.5 | 41.8 | 46.9 | 50.8 |
> | Qwen2.5-14B (Single) | 15.6 | 23.0 | 30.6 | 38.0 | 45.1 | 51.5 | 56.7 |
> | Ratio 2:1 (Favor Q3) | 16.6 | 23.3 | 30.7 | 38.0 | 44.6 | 50.9 | 57.5 |
> | Ratio 1:1 (Balanced) | 16.7 | 23.7 | 31.6 | 39.6 | 47.0 | 53.9 | 60.0 |
> | **Ratio 1:2 (Favor Q2.5)** | **16.8** | **24.0** | **31.8** | **39.9** | **47.9** | **55.4** | **62.5** |
>
> Results show that performance gains are robust across this range. In addition, favoring the better-performing model results in even better result. This aligns with our design in the main experiments, where current policy is favored over stale policy.
>
>
>
> ### Hyperparameter in RL
>
> Due to the prohibitive computational costs of our large-scale setting and strict rebuttal time constraints, we were unable to perform an exhaustive hyperparameter sweep. However, to serve as a sanity check, we conducted a targeted sensitivity experiment. The results confirm that our current configuration is effective and that the framework exhibits a reasonable degree of robustness to hyperparameter variations.
>
>
>
>
>
>
>
> Specifically, we changed KL loss coefficient from 3e-4 to 1e-3, learning rate from 1e-6 to 3e-7, incomplete punishment from -0.2 to -1.0. Emperically, these hyperparameters are most sensitive.
>
> | Method / Training pass rate | Step 20 | Step 40 | Step 60 | Step 80 | Step 100 | Step 120 | Step 140 |
> |---|---|---|---|---|---|---|---|
> | current hyp. | 0.29 | 0.34 | 0.40 | 0.44 | 0.47 | 0.49 | 0.52 |
> | other hyp. | 0.29 | 0.31 | 0.37 | 0.41 | 0.44 | 0.47 | 0.50 |
>
> The training curve under alternative hyperparameters exhibits a consistent upward trend that closely parallels the optimal configuration. While there is a persistent marginal gap of approximately 3% in pass rate—roughly equivalent to a lag of 20 training steps—the trajectory confirms that the model continues to learn stably (also shown in Figure 10, Appendix D.5.2). This indicates that while hyperparameters may influence the rate of convergence, **the framework is robust to hyperparameters and the overall training stability remains unaffected**.
>
> While this experiment does not definitively prove that our current hyperparameters are optimal, we believe it suggests that our selection is at least near a local optimum.
>
>
>
> If the reviewer deems a broader search critical for acceptance, we are fully committed to dedicating the necessary resources to conduct and include a more comprehensive analysis in the final version of the paper.

---

> ### Author Response · Authors · 2025-11-23
>
> ## Task Advantage Normalization & Reward Design
>
> > Q2 : Can the authors provide more granular per‑task and per‑instance breakdowns of reward scaling and failure modes to better demonstrate the stability of TAN?
>
> To address the request for further evidence of TAN's stability, we have conctucted an experiment on another set of hyperparameter.
>
>
> Specifically, in the sub-optimal hyperparameter setting mentioned above, we compare the training pass rate curve with and without TAN.
>
>
> | Method / Training pass rate | Step 20 | Step 40 | Step 60 | Step 80 | Step 100 | Step 120 | Step 140 |
> |---|---|---|---|---|---|---|---|
> | w/ task adv. norm | 0.29 | 0.31 | 0.37 | 0.41 | 0.44 | 0.47 | 0.50 |
> | w/o task adv. norm | 0.28 | 0.31 | 0.36 | 0.40 | 0.42 | 0.43 | 0.44 |
>
> The baseline without TAN rapidly converges to a suboptimal plateau, while the training curve with TAN maintains a continuous upward trend throughout the session (also shown in Figure 11b, Appendix D.5.3). This granular view confirms that TAN effectively stabilizes the learning process and prevents specific task rewards or failure modes from dominating the gradient updates.
>
>
>
> > Q3 : How sensitive is AGENTRL to choice of reward normalization and environment design? Is there any failure cases if task heterogeneity is heightened further?
>
>
> We adopted the $[0, 1]$ normalization simply as a standard convention for defining success probabilities, without extensive tuning. **Theoretically**, our TAN mechanism along with the normalization in advantage estimators (e.g. GRPO) renders the framework insensitive to such design choices. Due to high computation cost, we were unable to conduct a thorough experiment. If the reviewer insists, we are open to conduct further experiments in the next few days.

---

> > ### Comment · Reviewer_rknx · 2025-11-25
> > **Thanks for the response**
> >
> > I appreciate the authors for their response. Most of my concerns are addressed. good luck

---

> > > ### Author Response · Authors · 2025-11-25
> > >
> > > We sincerely thank you for the positive feedback! We are glad to hear that our response has addressed your concerns. If you have any further questions or if there is anything further we can do to merit a higher score, please let us know.

---

### Official Review · Reviewer_DsdZ · 2025-11-01

**Soundness:** 2
**Presentation:** 2
**Contribution:** 3
**Rating:** 4
**Confidence:** 3

**Summary:**

This paper presents AGENTRL, a scalable framework for multi-turn and multi-task reinforcement learning of large language model agents. It introduces an asynchronous rollout–training pipeline, a unified function-call–based environment API, and novel algorithmic components such as cross-policy sampling and task advantage normalization, achieving state-of-the-art performance across five agentic benchmarks and demonstrating strong generalization to unseen tasks

**Strengths:**

- The asynchronous multi-turn training framework effectively eliminates the inefficiency of synchronous rollouts, reducing GPU idle time and improving overall throughput while maintaining stable policy updates.
  - The scalable multi-task environment infrastructure provides a unified function-call–based API and containerized deployment with centralized control, enabling efficient orchestration and management of heterogeneous environments at scale.
  - The algorithmic innovations, including cross-policy sampling and task advantage normalization, significantly enhance exploration and stabilize multi-task reinforcement learning, resulting in consistent performance gains across benchmarks.

**Weaknesses:**

- The proposed asynchronous generation-training architecture inevitably introduces a significant off-policy problem, where training data is generated by stale policies. However, for a framework based on an on-policy algorithm , AGENTRL fails to mention any mechanism to correct for the bias caused by this policy lag. This oversight of a core theoretical challenge leaves the stability and convergence of its training process theoretically unsubstantiated and makes the description of its asynchronous design ambiguous.
- The motivation and necessity of Cross-Policy Sampling are not sufficiently justified. This method, which samples actions from historical models, directly conflicts with the assumptions of on-policy algorithms and introduces uncorrected distribution shift risks. Furthermore, the paper links its motivation to solving "model collapse," yet provides no evidence that such collapse actually occurs. More critically, the experiments lack a comparison with simpler alternatives, such as standard experience replay, making it impossible to determine whether the complexity of this design is truly necessary or if its claimed benefits surpass those of established techniques
- Task Advantage Normalization (TAN) sounds complex but offers limited practical benefits. At its core, it simply performs z-score normalization independently for each task — a common trick that lacks higher-level insight.

**Questions:**

- Given the off-policy data introduced by asynchronous training, does your framework include any mechanisms (even implicit ones) to mitigate its negative impact on the on-policy algorithm (GRPO)?
- Regarding Cross-Policy Sampling, how do you theoretically justify that training directly on samples from older policies is stable and unbiased without applying off-policy corrections?
- Have you conducted experiments comparing Cross-Policy Sampling with standard experience replay? If not, what unique and irreplaceable advantages do you believe the former holds over the latter?

---

> ### Author Response · Authors · 2025-11-23
>
> We sincerely thank Reviewer DsdZ for their constructive feedback. We appreciate their recognition of the effectiveness and scalability of our framework, as well as the performance gains achieved. We address the raised concerns and questions below.
>
> ## Async Training and Off-Policy Bias
> > W1 : The proposed asynchronous generation-training architecture inevitably introduces a significant off-policy problem, where training data is generated by stale policies. However, for a framework based on an on-policy algorithm , AGENTRL fails to mention any mechanism to correct for the bias caused by this policy lag. This oversight of a core theoretical challenge leaves the stability and convergence of its training process theoretically unsubstantiated and makes the description of its asynchronous design ambiguous.
>
> > Q1: Given the off-policy data introduced by asynchronous training, does your framework include any mechanisms (even implicit ones) to mitigate its negative impact on the on-policy algorithm (GRPO)?
>
> > Q2: Regarding Cross-Policy Sampling, how do you theoretically justify that training directly on samples from older policies is stable and unbiased without applying off-policy corrections?
>
>
>
>
>
>
> ### Comparison with Synchronous Pipeline
>
> To empirically validate the impact of off-policy bias, we conducted a new comparative experiment between our asynchronous pipeline and a synchronous baseline.
>
> | Method / Training pass rate | Step 20 | Step 40 | Step 60 | Step 80 | Step 100 | Step 120 | Step 140 |
> |---|---|---|---|---|---|---|---|
> | sync training | 0.49 | 0.51 | 0.56 | 0.60 | 0.63 | 0.65 | 0.67 |
> | async training | 0.48 | 0.51 | 0.55 | 0.58 | 0.62 | 0.66 | 0.69 |
>
> **The experiment results show that the training curves are nearly identical**, confirming that the off-policy bias introduced by our asynchronous design has a **negligible impact on training convergence and stability** (also see Appendix D.5.1 Figure 9). This allows AgentRL to achieve the massive efficiency gains demonstrated in the main paper without compromising the convergence quality.
>
> ### Mechanisms to Mitigate Off-Policy Bias
>
> Theoretically, GRPO handles this via Importance Sampling. This **internal importance sampling mechanism effectively neutralizes the off-policy bias**, ensuring accurate updates. This mechanism corrects off-policy bias introduced by both asynchronous pipeline and the cross-policy sampling. To ensure the correction is exact, we **directly use the tokens and logprobs from the rollout engine**(which is exactly $\pi_{old}$) instead of re-computing them.
>
> In addition, to avoid data from getting too far away from current policy, we enforce that the data queue to be non-accumulative. At each step, all data in the queue will be cleared from the queue and fed to the training engine.

---

> ### Author Response · Authors · 2025-11-23
>
> ## about Cross Policy Sampling
>
> > W2: The motivation and necessity of Cross-Policy Sampling are not sufficiently justified. This method, which samples actions from historical models, directly conflicts with the assumptions of on-policy algorithms and introduces uncorrected distribution shift risks. Furthermore, the paper links its motivation to solving "model collapse," yet provides no evidence that such collapse actually occurs. More critically, the experiments lack a comparison with simpler alternatives, such as standard experience replay, making it impossible to determine whether the complexity of this design is truly necessary or if its claimed benefits surpass those of established techniques
>
> > Q3: Have you conducted experiments comparing Cross-Policy Sampling with standard experience replay? If not, what unique and irreplaceable advantages do you believe the former holds over the latter?
>
> ### Motivation & "Model Collapse" Evidence
>
> We adopted cross-policy sampling to encourage model exploration. We discovered that cross-policy sampling strategy effectively broaders the ability range of models in offline sampling. This has already been demonstrated in the paper, we list the data below. Under this observation, we apply cross-policy sampling in RL to encourage exploration.
>
> | Model | P@1 | P@2 | P@4 | P@8 | P@16 | P@32 | P@64 |
> | :--- | :---: | :---: | :---: | :---: | :---: | :---: | :---: |
> | Llama-3.1-8B (Single) | 11.4 | 16.6 | 21.4 | 26.0 | 30.9 | 36.2 | 41.9 |
> | Qwen2.5-14B (Single) | **18.7** | **26.2** | **33.2** | 39.8 | 46.2 | 51.9 | 56.8 |
> | **Cross-Policy (Qwen + Llama)** | 16.6 | 24.2 | 32.0 | **40.2** | **48.7** | **56.8** | **63.7** |
>
> Also, we would like to clarify that we initially adopted the term "model collapse" from the cited literature that describes the variance reduction observed when training on self-generated data. We found this analogous to the exploration decay in our RL setting: As training progresses, the policy tends to lose exploration capability and converges prematurely to a local optimum. This effect is already demonstrated in our ablation study.
>
> To further demonstrate this, we have conducted an experiment on DB environment to further demonstrate the phenomenon and the effect of our algorithm.
>
> | Method / Training pass rate | Step 40 | Step 80 | Step 120 | Step 160 | Step 200 | Step 240 | Step 280 | Step 320 |
> |---|---|---|---|---|---|---|---|---|
> | w/ cross policy sampling | 0.49 | 0.57 | 0.74 | 0.79 | 0.82 | 0.84 | 0.85 | 0.85 |
> | w/o cross policy sampling | 0.51 | 0.58 | 0.66 | 0.70 | 0.73 | 0.74 | 0.76 | 0.77 |
>
> As shown in the result, with and without cross-policy sampling have substantial difference on training curve, especially when the training progresses (also shown in Figure 11a, Appendix D.5.3). **This demonstrates that our method can effectively improve model's exploration.**
>
> To avoid potential confusion, we have removed the term "model collapse" from the paper. We have also updated the section to provide a more detailed and precise explanation of the specific exploration challenges that cross-policy sampling addresses.
>
> ### Comparison with Experience Replay
>
> Theoretically, cross-policy sampling differs fundamentally from experience replay. While experience replay functions as a data reuse mechanism that replays static, offline trajectories, **cross-policy sampling acts as an active exploration mechanism** that generates new, dynamically consistent trajectories. By mixing actions from recent policy snapshots during online generation, cross-policy sampling injects necessary diversity to counteract exploration decay.
>
> Concretely, we validated this unique advantage through a new comparative experiment requested by the reviewer.
>
> | Method / Training pass rate | Step 40 | Step 80 | Step 120 | Step 160 | Step 200 | Step 240 |
> |---|---|---|---|---|---|---|
> | w/ cross policy sampling | 0.49 | 0.57 | 0.74 | 0.79 | 0.82 | 0.84 |
> | w/ experience replay | 0.54 | 0.62 | 0.72 | 0.75 | 0.78 | 0.79 |
>
> The results show that **cross-policy sampling maintains a clear and substantial performance margin over experience replay** throughout training (also shown in Figure 12, Appendix D.5.4). This confirms that cross-policy sampling offers **irreplaceable benefits in sustaining exploration** that cannot be replicated by simple data reuse, fully justifying its necessity.

---

> ### Author Response · Authors · 2025-11-23
>
> ## Task Advantage Normalization
>
> > W3 : Task Advantage Normalization (TAN) sounds complex but offers limited practical benefits. At its core, it simply performs z-score normalization independently for each task — a common trick that lacks higher-level insight.
>
>
>
> ### Regarding Practical Benefit
>
> Regarding practical benefits, our ablation study already demonstrate a **substantial gain (59.4% → 65.0%, Table 6)** when applying TAN. In addition, TAN also provides **crucial algorithmic robustness** in sub-optimal settings. To demonstrate this, we conducted an additional experiment using sub-optimal hyperparameters.
>
> | Method / Training pass rate | Step 20 | Step 40 | Step 60 | Step 80 | Step 100 | Step 120 | Step 140 |
> |---|---|---|---|---|---|---|---|
> | w/ task adv. norm | 0.29 | 0.31 | 0.37 | 0.41 | 0.44 | 0.47 | **0.50 (+13.6%)** |
> | w/o task adv. norm | 0.28 | 0.31 | 0.36 | 0.40 | 0.42 | 0.43 | 0.44 |
>
> In this harsher setting, the effect of TAN is pronounced: removing it leads to training instability and significantly lower success rates, whereas enabling TAN maintains consistent convergence and gains a substantial better result (relatively +13.6%) (also shown in Figure 11b, Appendix D.5.3). We have added these results to the paper to highlight the value of our method in ensuring stability across diverse conditions.
>
> ### Regarding higher-level Insight & Complexity
>
> We would like to emphasize that **our primary contribution and higher-level insight here lies in identifying the critical training imbalance in multi-task LLM training and solving it via normalization**, where tasks with different advantage statistics can possibly dominate model updates and destabilize learning. We demonstrate that applying this standard technique is an effective and scalable solution to this specific problem, allowing the model to balance diverse tasks without requiring manual reward engineering for each environment.
>
> Regarding the perceived complexity, we apologize if the formal notation in our initial submission made the method appear unnecessarily convoluted. In practice, TAN is intentionally designed to be **lightweight and easy to implement**, requiring only approximately 20 lines of code, as can be seen in our repository. We argue that **our method is simple yet effective, and does not affect training efficiency**. We have revised the corresponding section in the paper to simplify the mathematical presentation, focusing more on the intuition and implementation simplicity to ensure the core concept is immediately clear to readers.

---

### Official Review · Reviewer_vrk7 · 2025-11-01

**Soundness:** 3
**Presentation:** 4
**Contribution:** 3
**Rating:** 6
**Confidence:** 5

**Summary:**

This paper presents AgentRL, a comprehensive framework for scalable multi-turn, multi-task reinforcement learning (RL) training of large language model (LLM) agents. The work addresses significant challenges in agentic RL by introducing: (1) a fully asynchronous generation-training pipeline for efficient multi-turn RL, (2) a unified function-call based API interface with containerized environment deployment and centralized control, (3) cross-policy sampling to enhance exploration in multi-turn settings, and (4) task advantage normalization to stabilize multi-task training.The authors evaluate AgentRL on five diverse agentic tasks (ALFWorld, DB, KG, OS, WebShop) across multiple model scales (3B-72B parameters). Results demonstrate that AgentRL-trained models significantly outperform strong baselines including GPT-5, Claude-Sonnet-4, and DeepSeek-R1, achieving state-of-the-art performance with an average success rate of 70.4%. Notably, a single multi-task trained model matches the best performance of five task-specific models while showing promising generalization to unseen tasks (BFCL-v3 benchmark).

**Strengths:**

Presents AgentRL, a comprehensive framework for scalable multi-turn, multi-task reinforcement learning (RL) training of large language model (LLM) agents, addressing infrastructure, environment, and algorithmic challenges simultaneously.

Achieves impressive performance gains across all tested tasks and outperforms strong proprietary baselines (GPT-5, Claude-Sonnet-4).
The asynchronous pipeline achieves significant throughput improvements, making the approach practically viable at scale.

Successfully demonstrates that a single multi-task trained model can match the best performance of five task-specific models while maintaining strong generalization capabilities.

**Weaknesses:**

Limited Theoretical Analysis: Cross-policy sampling's impact on policy distribution is mentioned but not formally analyzed

Evaluation Scope: Limited diversity in task types (no long-horizon embodied tasks, no code generation with execution, no real-world web automation); OOD evaluation limited to one benchmark (BFCL-v3)

Cross-Policy Sampling Analysis: "Stale engines" update schedule unclear (every "multiple steps" is vague); Limited analysis of the exploration-exploitation tradeoff; Section 4.3.1's inference experiments use different models (Llama vs Qwen) but training uses same model - this asymmetry needs justification

Scalability and Cost: No discussion of computational costs or training time

**Questions:**

1. How exactly is the "stale engine" update frequency determined? Is it task-dependent or fixed? What is the probability distribution for sampling from different policies at each step? Have you experimented with more than 2 policies? How does performance scale with the number of policies?

2.What is the maximum queue size for the data buffer mentioned in Section 3.1? How much off-policy bias is introduced? Can you provide a comparison with synchronous training in terms of training time, convergence speed, and performance ceiling? How do you ensure the trajectories remain "as up-to-date as possible" when the queue can accumulate data?

3. Why normalize at the token level rather than the trajectory level? Have you experimented with other normalization strategies？Does this introduce any bias when tasks have very different token-length distributions?

4.The BFCL-v3 results show modest improvements on single-turn tasks.  Can you provide more analysis on what makes the model generalize to new tasks? Have you tested on tasks completely different from the training distribution?

5. Section 4.3.1 uses heterogeneous models (Llama vs Qwen) for inference experiments, but training uses temporal versions of the same model family. Does this mean cross-policy sampling during training merely augments offline data diversity rather than introducing fundamentally different exploration mechanisms?

---

> ### Author Response · Authors · 2025-11-23
>
> We sincerely thank reviewer vrk7 for their thorough review and positive assessment. We appreciate their recognition of AgentRL's contributions.
>
> ## about Cross Policy Sampling
>
> We appreciate the reviewer's detailed inquiries regarding cross-policy sampling. Before addressing specific questions, we summarize the key points regarding its effectiveness and soundness:
>
> * **Effectiveness:** Cross-policy sampling demonstrates consistent and substantial improvements, achieving a **+20.2% boost** in offline pass@k experiments and **+6.3% to +10.4% gains** in online RL training.
> * **Soundness:**
>     * We have addressed theoretical concerns by adding a formal analysis based on **valid manifolds**.
>     * We clarify that off-policy bias is strictly managed via a **non-accumulating data flow** and **importance sampling** using direct rollout logprobs.
>     * We conducted additional experiments with **intra-family models** and **multi-model pools**, resolving concerns regarding model asymmetry and confirming the positive scaling effect of cross-policy sampling.
>
>
>
> > W1: Limited Theoretical Analysis: Cross-policy sampling's impact on policy distribution is mentioned but not formally analyzed.
>
> The Cross-Policy Sampling strategy originated from an experimental observation rather than a formal theoretical derivation. Consequently, our initial submission did not emphasize a formal analysis in the main paper. Here we try to formalize this process and give some intuitive results.
>
> Formally, let $\mathcal{M} = \{\pi_{\theta_0}, \dots, \pi_{\theta_K}\}$ be the set of the current and historical policies. Unlike standard sampling from a single distribution, Cross-Policy Sampling (CPS) generates a trajectory $\tau_{cps}$ where each action $a_t$ is drawn via a two-stage process: first sampling a model index $k \sim \mathcal{U}(0, K)$, then sampling $a_t \sim \pi_{\theta_k}(\cdot | s_t)$.
>
> The theoretical advantage can be described via the valid solution space   $\mathcal{L}_{valid}$
> (linguistically coherent sequences).
>
> While a single policy $\pi_{\theta}$ tends to collapse to a specific mode, CPS approximates the union of the support of all history policies:
> \begin{equation}
>     \text{Support}(\tau_{cps}) \approx \bigcup_{\pi \in \mathcal{M}} \text{Support}(\tau_{\pi}) \cap \mathcal{L}_{valid}
> \end{equation}
> This formulation highlights why CPS is superior to simply increasing sampling temperature (entropy): high temperature expands exploration isotropically, often drifting into invalid regions (hallucinations), whereas CPS expands exploration strictly along the **valid manifolds** defined by previous coherent policies.
>
> We have added a deeper discussion on this topic in Appendix B.3.
>
> > W3:  Cross-Policy Sampling Analysis: "Stale engines" update schedule unclear (every "multiple steps" is vague); Limited analysis of the exploration-exploitation tradeoff;
>
> > Q1: How exactly is the "stale engine" update frequency determined? Is it task-dependent or fixed? What is the probability distribution for sampling from different policies at each step? Have you experimented with more than 2 policies? How does performance scale with the number of policies?
>
> The stale rollout engines synchronize weights every $T$ steps (a fixed hyperparameter, default $T=25$), and actions are sampled from these engines with probability $\rho$ (default $\rho=0.25$), this can be considered as a Bernoulli distribution parameterized by $\rho$.
>
> As stated above, cross-policy sampling is able to encourage exploration without discouraging exploitation. We demonstrate this via a comparative RL experiment on DB.
>
> | Method / Training pass rate | Step 40 | Step 80 | Step 120 | Step 160 | Step 200 | Step 240 | Step 280 | Step 320 |
> |---|---|---|---|---|---|---|---|---|
> | w/ cross policy sampling | 0.49 | 0.57 | 0.74 | 0.79 | 0.82 | 0.84 | 0.85 | **0.85(+10.4%)** |
> | w/o cross policy sampling | 0.51 | 0.58 | 0.66 | 0.70 | 0.73 | 0.74 | 0.76 | 0.77 |
>
> The result shows that with and without cross-policy sampling have substantial difference on training curve (+10.4% gain in 320 steps), especially when the training progresses (also shown in Figure 11a, Appendix D.5.3). **This demonstrates that our method has improved model's exploration without harming its exploitation.**

---

> ### Author Response · Authors · 2025-11-23
>
> While our primary training setup employs a dual-pool strategy (Latest + Stale), **we investigated the scaling potential through supplementary experiments mixing three distinct models** (e.g., Qwen2.5, Qwen3, GLM-4) on DB. The results are listed below:
>
> | Task | Model / Policy | P@1 | P@2 | P@4 | P@8 | P@16 | P@32 | P@64 |
> | :--- | :--- | :---: | :---: | :---: | :---: | :---: | :---: | :---: |
> | **DB** | GLM-4-9B | 3.2 | 5.6 | 9.6 | 15.5 | 22.6 | 29.6 | 36.1 |
> | | Qwen3-14B-Instruct | **53.9** | **60.3** | 64.3 | 67.4 | 70.0 | 72.0 | 73.5 |
> | | Qwen2.5-14B-Instruct | 48.6 | 58.7 | 64.1 | 67.0 | 69.3 | 71.6 | 74.0 |
> | | Cross (Qwen3 + GLM4) | 37.1 | 48.3 | 56.6 | 62.1 | 66.1 | 69.4 | 72.3 |
> | | Cross (Qwen2.5 + GLM4) | 27.3 | 41.7 | 54.9 | 63.2 | 67.3 | 69.8 | 71.3 |
> | | Cross (Qwen3 + Qwen2.5) | 48.4 | 57.5 | 63.0 | 66.5 | 68.9 | 71.1 | 73.7 |
> | | **Cross (All 3 Models)** | 49.6 | 59.6 | **65.5** | **69.0** | **71.5** | **73.7** | **75.7** |
>
>
> We observed that performance positively scales with the number of policies, confirming that Cross-Policy Sampling effectively aggregates diverse capabilities and can scale via adding policies. Despite this, we maintain dual-policy setting in the main experiment to simplify the framework implementation. This result is also added to Appendix D.6.3.
>
>
> > W3: Section 4.3.1's inference experiments use different models (Llama vs Qwen) but training uses same model - this asymmetry needs justification
>
> > Q5: Section 4.3.1 uses heterogeneous models (Llama vs Qwen) for inference experiments, but training uses temporal versions of the same model family.
>
> The inference experiment in Section 4.3 using distinct model families (Qwen vs. Llama) served as a preliminary proof-of-concept, intentionally designed to maximize policy divergence to clearly demonstrate how mixing policies expands the solution space. For the actual RL training, we adopted intra-family mixing (current vs. stale versions) to keep the policy divergence bounded, thereby avoiding the severe off-policy instability.
>
>
> Specifically, using heterogeneous models for Cross-Policy Sampling during training would introduce two primary issues. First, it would lead to **inconsistent token spaces** (i.e., different tokenizers). Maintaining token consistency is critical to prevent off-policy problem. Second, even if the token spaces could be aligned, this approach would likely cause an **excessively large off-policy divergence**, which would destabilize training.
>
> We verify the efficacy of mixing models from same family: Base Model and its RL-optimized version. We conduct experiments on WebShop. The cross-policy sampling results consistently outperform the single-model baseline, confirming the benefit of this strategy.
>
> | Task | Model / Policy | P@4 | P@8 | P@16 | P@32 | P@64 |
> | :--- | :---: | :---: | :---: | :---: | :---: | :---: |
> | **WebShop** |  Before RL (Qwen2.5-14B-Instruct) | 30.6 | 38.0 | 45.1 | 51.5 | 56.7 |
> | | After RL | 62.5 | 63.5 | 64.2 | 64.7 | 65.0 |
> | | **Cross(Before & After RL)** | **63.3** | **70.3** | **74.6** | **78.2** | **81.5** |
>
> Therefore, we adopted our current approach (using recent snapshots of the same model) as a deliberate trade-off. This method allows us to retain the exploratory benefits of cross-policy sampling while avoiding the instability risks associated with severe off-policy updates.
>
> > Q5: Does this mean cross-policy sampling during training merely augments offline data diversity rather than introducing fundamentally different exploration mechanisms?
>
>
> We interpret the reviewer's description of "augmenting offline data diversity" as being analogous to mechanisms like Experience Replay, where old generated data are stored and replayed to train current policy. To prove this distinction, **we conduct a comparative experiment against Experience Replay**.
>
> | Method / Training pass rate | Step 40 | Step 80 | Step 120 | Step 160 | Step 200 | Step 240 |
> |---|---|---|---|---|---|---|
> | w/ cross policy sampling | 0.49 | 0.57 | 0.74 | 0.79 | 0.82 | 0.84 |
> | w/ experience replay | 0.54 | 0.62 | 0.72 | 0.75 | 0.78 | 0.79 |
>
> The results demonstrate that CPS maintains a clear and substantial performance margin over ER throughout the training process (Also shown in Figure 12, Appendix D.5.4). This confirms that CPS functions as a fundamental exploration mechanism rather than simple data augmentation.

---

> ### Author Response · Authors · 2025-11-23
>
> ## about Eval Scope
>
> > W2 : Evaluation Scope: Limited diversity in task types (no long-horizon embodied tasks, no code generation with execution, no real-world web automation); OOD evaluation limited to one benchmark (BFCL-v3)
>
> > Q4 : The BFCL-v3 results show modest improvements on single-turn tasks. Can you provide more analysis on what makes the model generalize to new tasks? Have you tested on tasks completely different from the training distribution?
>
> ### Regarding Task Diversity:
>
> We respectfully submit that our current evaluation on AgentBench already provides a broad and comprehensive coverage of core agentic capabilities across diverse domains. Five environments (OS, DB, KG, Webshop, Alfworld) effectively represent key real-world applications: **coding and execution** (OS, DB), **retrieval-augmented generation (RAG) and reasoning** (KG), **web** interaction (WebShop), and **simulated embodied planning** (AlfWorld)."
>
> Furthermore, AgentRL is explicitly architected for extensibility. We view this work as a foundational prerequisite that paves the way for scaling agent reinforcement learning to an even wider array of complex domains in the future.
>
> Also, we emphasize that our OOD benchmark, BFCL-v3, offers similarly broad coverage. It encompasses multiple distinct subtasks and effectively functions as an aggregation of several sub-benchmarks. This inherent richness ensures that the evaluation is sufficiently diverse to rigorously test generalization.
>
>
> ### Regarding the Performance on BFCL-v3:
> The modest gains on single-turn tasks are expected. Our RL training explicitly optimizes for **complex multi-turn interactions**, enhancing sequential planning and context tracking. Consequently, while we observe significant gains in multi-turn scenarios, single-turn performance relies primarily on the base model's pre-established instruction-following capabilities (from SFT), resulting in smaller marginal gains from our interaction-centric RL. We have added this analysis to the paper.
>
> ### Regarding the Generalization Analysis:
> We maintain that BFCL-v3 serves as a rigorous OOD benchmark. The domains included are **structurally and semantically different from our training environments**. Therefore, the results in the paper effectively demonstrate that our method confers robust generalization capabilities to **completely unseen task distributions**.
>
> ## about Training Details
>
> > Q2 : What is the maximum queue size for the data buffer mentioned in Section 3.1? How much off-policy bias is introduced? Can you provide a comparison with synchronous training in terms of training time, convergence speed, and performance ceiling? How do you ensure the trajectories remain "as up-to-date as possible" when the queue can accumulate data?
>
> ### Queue Size and Off-Policy Bias & Ensuring data freshness:
> The **data buffer is capped at 2x the configured per-step batch size**. To bound off-policy bias, we enforce the queue to be fully drained at every training step, ensuring that **data won't accumulate**.
>
>
> ### Comparison with Synchronous Training:
>
> As shown in Figure 4, the asynchronous pipeline achieves nearly **$2\times$ higher throughput** than the synchronous baseline. To strictly verify that this efficiency does not come at the cost of stability, convergence speed, or performance ceiling, we conducted a new controlled comparative experiment on the DB environment.
>
> | Method / Training pass rate | Step 20 | Step 40 | Step 60 | Step 80 | Step 100 | Step 120 | Step 140 |
> |---|---|---|---|---|---|---|---|
> | sync training | 0.49 | 0.51 | 0.56 | 0.60 | 0.63 | 0.65 | 0.67 |
> | async training | 0.48 | 0.51 | 0.55 | 0.58 | 0.62 | 0.66 | 0.69 |
>
> The results demonstrate that the training curves for synchronous and asynchronous modes are nearly identical (average difference 1.49%) (also shown in Figure 9, Appendix D.5.1). This confirms that the off-policy bias introduced by **the asynchronous mechanism has a negligible impact on convergence and final performance**, while retaining the substantial advantage in wall-clock training speed.
>
> > W4: Scalability and Cost: No discussion of computational costs or training time
>
> We have added a discussion on computational costs in Appendix D.7. All experiments were conducted on a cluster of 8 gpus/node * 4 nodes (32 GPUs total, each with ~1500 TFLOPs BF16 peak performance).
>
> * **14B Model (1500 steps):** Completed in **~59 hours** wall-clock time (~ 1888 GPU hours).
> * **32B Model (1500 steps):** Completed in **~101 hours** wall-clock time (~ 3232 GPU hours).
>
> These numbers include all overheads, such as computational costs incurred by training resumption. The successful scaling to a 32-GPU cluster with reasonable training durations demonstrates the framework's scalability and efficiency.

---

> ### Author Response · Authors · 2025-11-23
>
> ## about Task Advantage Normalization
>
> > Q3: Why normalize at the token level rather than the trajectory level? Have you experimented with other normalization strategies？Does this introduce any bias when tasks have very different token-length distributions?
>
> ### Why Token Level
>
> We employ token-level normalization because the underlying optimization objective aggregates per-token log-probabilities. Normalizing at this granularity ensures that the gradient contribution of each decision step is standardized, thereby preventing tasks with longer horizons from disproportionately dominating the update solely due to sequence length.
>
> On the other hand, normalizing advantage at the trajectory level will have no effect when incorprating with GRPO algorithm. As the advantage is already normalized by the GRPO at trajectory level.
>
> To further demonstrate the effectiveness of TAN, **we conducted an additional experiment using suboptimal hyperparameters**.
> | Method / Training pass rate | Step 20 | Step 40 | Step 60 | Step 80 | Step 100 | Step 120 | Step 140 |
> |---|---|---|---|---|---|---|---|
> | w/ task adv. norm | 0.29 | 0.31 | 0.37 | 0.41 | 0.44 | 0.47 | 0.50 |
> | w/o task adv. norm | 0.28 | 0.31 | 0.36 | 0.40 | 0.42 | 0.43 | 0.44 |
>
> When TAN is removed, the training curve quickly stagnates and converges to a suboptimal plateau (also shown in Figure 11b, Appendix D.5.3). This suggests that without the normalization provided by TAN, the optimization process becomes dominated by unbalanced reward scales, trapping the policy in a local optimum, whereas TAN effectively balances these signals to ensure continuous improvement.
>
> ### Length Bias
>
> Regarding the length bias problem, **our five tasks already vary from turns and total response length greatly**. For example, when converged, os takes averagely 2 turns while alfworld takes more than ten turns. The total response length varies from <100 tokens to more than 6k tokens throughout the training. Under this condition, Our addtional experiment and original ablation studies in the paper is sufficient to prove that TAN effectively improves training efficiency and stability, without introducing length bias.

---

### Author Response · Authors · 2025-12-03

Dear Area Chair,

We are writing to provide a brief summary of our rebuttal process.

### Consensus on Strengths

Across the board, reviewers consistently recognized the following strengths in our work:
* **Scalable Infrastructure:** Reviewers highlighted the practical value of our fully asynchronous generation-training pipeline and unified environment interface for large-scale agent training.
* **Strong Performance:** All reviewers acknowledged the impressive performance gains, noting that our method outperforms strong proprietary baselines (e.g., GPT-5, Claude-Sonnet-4) and achieves state-of-the-art results on diverse benchmarks.
* **Algorithmic Innovation:** The proposed Cross-Policy Sampling and Task Advantage Normalization were recognized as practical and well-motivated contributions for stabilizing multi-task, multi-turn RL.

### Reviewers' Concerns

We have successfully addressed the questions from Reviewers 3aoe and rknx. Notably, **Reviewer 3aoe increased their rating from 6 to 8**, and Reviewer rknx confirmed that most of their concerns have been resolved.

Regarding **Reviewer DsdZ (Score 4)**, while they have not yet responded, we have addressed all their concerns with new controlled experiments:

* **Concern on Asynchronous Bias:** The reviewer worried that asynchronous training would introduce instability.
    **Response:** We conducted a new comparison between Asynchronous and Synchronous training. The results show the training curves are nearly identical (average difference **< 1.49%**), proving the off-policy bias is negligible.
* **Concern on Cross-Policy Sampling:** The reviewer questioned its necessity compared to standard methods.
    **Response:** We ran a new ablation study comparing Cross-Policy Sampling against standard Experience Replay. Cross-Policy Sampling consistently outperforms Experience Replay (e.g., pass rate **0.84** vs **0.79** at step 240), confirming it provides essential exploration that simple data reuse cannot achieve.
* **Concern on Task Advantage Normalization:** The reviewer questioned its practical utility.
    **Response:** We first pointed out that our original ablation study already demonstrated a substantial gain (**59.4%** vs **65.0%**). In addition, we conducted new experiments under sub-optimal hyperparameter settings. The results show that removing Task Advantage Normalization leads to training failure, whereas including it secures convergence and achieves a significant gain (**+13.6%** relative success rate).

Regarding **Reviewer vrk7 (Score 6)**, we also provided extensive additional results to address their detailed inquiries:

* **Concern on Cross-Policy Sampling Effectiveness:** The reviewer asked for analysis on the exploration-exploitation tradeoff.
    **Response:** We provided a comparative RL experiment on the DB task. The results demonstrate that Cross-Policy Sampling significantly boosts performance (**0.85**) compared to the baseline without it (**0.77** at step 320), proving it improves exploration without harming exploitation.
* **Concern on Scaling with Multiple Policies:** The reviewer asked if performance scales with the number of policies.
    **Response:** We conducted experiments mixing three distinct models. The results confirmed that performance positively scales with the number of policies (e.g., mixing 3 models achieved **75.7** vs the best single model **73.5**).
* **Concern on Inference vs. Training Discrepancy:** The reviewer asked about the justification for using intra-family models during training.
    **Response:** We validated that mixing models from the same family (Base Model + RL version) yields consistent gains (e.g., **81.5** vs **56.7** on WebShop), justifying our design choice to avoid severe off-policy instability while retaining exploration benefits.

We hope this summary assists in your final assessment.

Best regards,
Authors

---

### Meta-Review · Area_Chair_wHkN · 2026-01-06

**Summary:**

The submission presents a strong and carefully engineered systems framework, but multiple reviewers expressed a recurring concern that the work’s primary contributions lie in orchestration and stabilization rather than in introducing a fundamentally new learning principle. In particular, reviewers questioned whether the key algorithmic components—cross-policy sampling and task advantage normalization—constitute novel ideas or well-motivated refinements of established techniques such as policy mixing for exploration and per-task normalization for multi-task stability. While the rebuttal convincingly addresses questions of soundness, robustness, and necessity (e.g., comparisons to experience replay, sync vs. async training, and ablations under noise and sub-optimal hyperparameters), these additions largely strengthen the case that the system works as intended rather than clarifying what is conceptually new relative to prior multi-task RL and agent-training pipelines. As noted by reviewers such as QRfN and sPAZ, the overall method can be read as a careful integration of known ingredients—self-generated trajectories, replay, normalization, and asynchronous execution—applied effectively at scale, but without a clear new objective, theoretical insight, or paradigm shift. In this sense, despite impressive empirical results and a thorough rebuttal, the work remains closer to an incremental systems advance than to a contribution that redefines the space, which leaves the novelty bar insufficiently cleared for acceptance.

**Reviewer Concerns:**

See above.

**Reviewer Scores:**

See above.

---

### Decision · Program_Chairs · 2026-01-26

Reject